

# Constraining fossil fuel CO₂ emissions from urban area using OCO-2 observations of total column CO₂

Xinxin Ye[1], Thomas Lauvaux[1], Eric A. Kort[2], Tomohiro Oda[3], Sha Feng[1], John C. Lin[4], Emily Yang[2], Dien Wu[4]

[1]Department of Meteorology and Atmospheric Science, The Pennsylvania State University, University Park, 16803, USA; [2]Department of Climate and Space Sciences and Engineering, University of Michigan, Ann Arbor, 48109, USA; [3]Goddard Earth Sciences Technology and Research, Universities Space Research Association/Global Modeling and Assimilation Office, NASA Goddard Space Flight Center, Greenbelt, 20706, USA; [4] Department of Atmospheric Sciences, University of Utah, Salt Lake City, 84132, USA

*Correspondence to*: Xinxin Ye (xxin.ye@gmail.com)

**Abstract.** Expanding urban populations and the significant contribution of cities to global fossil-fuel $CO_2$ ($CO_{2ff}$) emissions emphasize the necessity of achieving independent and accurate quantifications of the emissions from urban area. In this paper, we assess the utility of total column dry air $CO_2$ mole fraction ($X_{CO2}$) data retrieved from NASA's Orbiting Carbon Observatory 2 (OCO-2) observations to quantify $CO_{2ff}$ emissions from cities. Observing System Simulation Experiments
(OSSEs) are implemented by forward modeling of meteorological fields and column $X_{CO2}$. The impact of transport model errors on the inverse emissions estimates is examined for two "plume cities" (Riyadh, Cairo) and a "basin city" (Los Angeles metropolitan region, LA). The pseudo data experiments indicate convergence of emission uncertainties related to transport model errors with increasing amount of observations. The 1-σ uncertainty of emission estimates is constrained to approximately 15%/5% with about 10 pseudo tracks for plume city/basin city. The systematic wind speed biases in simulated
wind fields for LA lead to overestimations in total $CO_{2ff}$ emission, which require data assimilation to improve high-resolution atmospheric transport. The contribution of biogenic fluxes gradients in urban and rural area of Pearl River Delta metropolitan region in China are examined by simulations with biospheric fluxes imposed by the Net Ecosystem Exchange (NEE) from multiple terrestrial biosphere models, which show about 24±21% (1σ) and 19±15% (1σ) contributions to the total $X_{CO2}$ enhancements for the two cases examined. The representations of transport model errors for the emission optimization are
examined for Riyadh, Cairo and LA in real cases. The determination of background $X_{CO2}$ is discussed for LA by using constant and simulated background with biospheric fluxes included, demonstrating the need of careful consideration of the variations in background $X_{CO2}$ for identifying concentration enhancements due to fossil fuel emissions.

## 1 Introduction

Since the preindustrial era the global atmospheric $CO_2$ concentration has increased by more than 40%, predominantly due
to $CO_2$ emissions from combustion of fossil fuels (coal, oil, and gas) and net land use change (Andres et al., 2012; Ciais et al., 2013; Rotty, 1983). Given that $CO_2$ is the most important greenhouse gas (GHG), the international community has pursued treaties and agreements on the mitigation and management of anthropogenic $CO_2$ emissions, e.g. the Kyoto Protocol (United





Nations, 1998) and the Paris Agreement (UNFCCC, 2015). Cities across the globe, where more than half of the world's population resides in, are the major sources of anthropogenic GHG emissions, accounting for more than 70% of the global energy-related $CO_2$ emissions (International Energy Agency, 2008). Although fossil fuel emissions are well quantified at national or regional level via emission inventory compilation, i.e. *bottom-up* approach, emissions at subnational or local level

are often based on disaggregation of national or regional/state emissions (Janssens-Maenhout et al., 2012; Kurokawa et al., 2013; Oda and Maksyutov, 2011), with a few exception such as (Gurney et al., 2012). For a limited number of cities, emissions are reported on voluntary basis or under climate action activities, such as the Global Covenant of Mayors (http://www.globalcovenantofmayors.org/), but the data are not spatial explicit and are often incomplete or rarely checked against scientific standards and procedures (Hutyra et al., 2014). Additionally, due to the differences in emission

disaggregation methods, *bottom-up* emissions estimates exhibit large discrepancy especially at high spatial and temporal resolutions (Ackerman and Sundquist, 2008; Gurney et al., 2012; Hogue et al., 2016; Oda et al., 2017b; Oda and Maksyutov, 2011; Turnbull et al., 2011). Thus independent, comprehensive and comparable emissions estimates are needed to enable transparent verification and monitoring of fossil-fuel $CO_2$ ($CO_{2ff}$) emissions from urban areas (Duren and Miller, 2012; Gurney et al., 2015; National Research Council, 2010).

Inverse modeling, or *top-down* approach assimilate atmospheric $CO_2$ observations with atmospheric transport modeling to quantify $CO_2$ fluxes from oceans and continents, as part of the "atmospheric inversions" (e.g. Bousquet, 2000; Ciais et al., 2010) Enting, 1998. It has been applied most often on global scale carbon fluxes at fairly coarse resolutions (Denning et al., 1995; Engelen et al., 2009; Gurney et al., 2002; Takagi et al., 2011). More recently, urban-scale $CO_{2ff}$ emissions have been constrained by inversion method by using ground-based measurements of $CO_2$ concentrations (Lauvaux et al., 2016; McKain et

al., 2012; Wunch et al., 2009) or by aircraft mass-balance (Cambaliza et al., 2014). However, only a handful of cities, mostly in Europe and North America, have been instrumented with networks of GHG sensors (Bréon et al., 2015; Davis et al., 2017; McKain et al., 2012; Miles et al., 2017; Verhulst et al., 2017). By comparison, space-based satellite retrievals of total column averaged dry air $CO_2$ mole fraction ($X_{CO2}$) featured with global spatial coverage are expected to facilitate the quantification and monitoring of $CO_{2ff}$ emissions from a significant number of cities (Duren and Miller, 2012; Kort et al., 2012; McKain et al.,

2012; Schneising et al., 2013). Many efforts have been made to relate remotely observed $X_{CO2}$ to $CO_{2ff}$ emissions on regional or local scales. For instance, $X_{CO2}$ enhancements are detected over industrial regions in Germany by the Scanning Imaging Absorption Spectrometer for Atmospheric Chartography (SCIAMACHY) on ENVISAT (Bovensmann et al., 1999; Buchwitz et al., 2005; Schneising et al., 2008, 2013). Using observations from the Greenhouse gases Observing SATellite (GOSAT) (Morino et al., 2011), discernible signatures of $CO_{2ff}$ emissions have been derived as regional $X_{CO2}$ contrasts between emission

and background regions (Janardanan et al., 2016; Keppel-Aleks et al., 2013). $X_{CO2}$ enhancements over megacities are identified by differencing measurements over urban and nearby background area (Kort et al., 2012). These enhancements in $X_{CO2}$ over or downwind of emission sources can be exploited as independent and observational constraints of $CO_{2ff}$ emissions via atmospheric inversion frameworks (Pillai et al., 2016).

    NASA's Orbiting Carbon Observatory 2 (OCO-2) (Crisp, 2008; Crisp et al., 2004) pioneered the contiguous high-

resolution mapping of global $CO_2$ concentrations. Since its launch on July 2, 2014, OCO-2 has been collecting spectroscopic observations of reflected sunlight in near infrared $CO_2$ and $O_2$ bands near midday (Crisp, 2015). The high spectral resolution and signal-to-noise ratio of the sensor enable an unprecedented 1 ppm retrieval error in $X_{CO2}$ (Eldering et al., 2017; Wunch et





al., 2017). The small nadir footprints of approximately 1.25 km $\times$ 2.25 km maximize probability of cloud-free observations in the presence of patchy clouds, which allow us to record concentration enhancement due to localized anthropogenic sources. OCO-2 data have been used to quantify $CO_{2ff}$ emissions on regional scale by Hakkarainen et al. (2016). However, the capability to improve estimates of anthropogenic carbon emissions from urban area has not been addressed, which will be

presented in this paper.

With high-density retrievals along repeated tracks near urban centers come several key challenges limiting our ability to constrain $CO_{2ff}$ emissions. Firstly, discernible $CO_{2ff}$ emission imprints can be limited due to the contamination by clouds and aerosols and limitations of spatial-temporal sampling coverage for local sources related to the revisit cycle of sun-synchronize polar orbit and the narrow tracks (e.g. 10.6 km at nadir of OCO-2). Secondly, atmospheric transport modeling errors and the

error correlations has been identified as a major source of uncertainty in the *top-down* method (Houweling et al., 2010; Lauvaux et al., 2012; Lauvaux and Davis, 2014; Lin and Gerbig, 2005; Miller et al., 2015). The improvement of model physics and/or data assimilation is required to reduce the systematic and random errors in the simulated meteorological conditions and to ensure policy-level requirements on annual emissions (less than 10% error) (Deng et al., 2017). Thirdly, the spatial heterogeneity of urban sources and existence of intense point sources (e.g. power plants) generate complex structures in urban

$CO_{2ff}$ emissions and hence the atmospheric $CO_2$ plumes (Feng et al., 2016). In order to obtain robust emissions estimates on urban scale, atmospheric transport models and prior emissions at high resolution are needed to capture variability in $X_{CO2}$ enhancements. Fourthly, varying $X_{CO2}$ background concentrations obscure the extraction of $CO_{2ff}$ signatures and attribution to local sources (Keppel-Aleks et al., 2013; Turnbull et al., 2016), since the background $X_{CO2}$ is influenced by complex interplay of biogenic fluxes, synoptic and dynamic processes (e.g. frontal passage), leading to diurnal and seasonal variability.

Additionally, the discrepancy of biogenic fluxes within and around cities in vegetated areas in mid-latitude, tropical and equatorial climates would contribute to local $X_{CO2}$ enhancements derived from observations along satellite tracks. Variations in biogenic fluxes and amplified plant physiology due to human interventions in cities (Hutyra et al., 2014) further complicate the interpretation of $CO_{2ff}$ signatures and source attribution.

The major goal of this paper is to evaluate the potential of OCO-2 $X_{CO2}$ observations along-tracks in nadir and glint modes

on the quantification of $CO_{2ff}$ emissions from cities. Since various local factors such meteorology, vegetation, and topography create city-specific difficulties in analyzing $X_{CO2}$ enhancements to quantify $CO_{2ff}$ emissions, we focus on a handful of cities with different typical features in $X_{CO2ff}$ enhancements. First, the $CO_{2ff}$ emissions from Riyadh, Saudi Arabia and Cairo, Egypt with populations of 6.2 million and 18.4 million are investigated to represent cities exhibiting $CO_{2ff}$ plumes dominated by atmospheric transport, referred to as *plume cities* hereafter. Both cities are located in moderate terrain variability with small

biogenic influence on background, and OCO-2 observations are available along several tracks. Second, the Los Angeles metropolitan area, the United States with a population of more than 13 million is selected to investigate urban emitters located in basins, featuring with strong enhancements in $X_{CO2}$ due to confined air masses and reduced advection/diffusion, referred to as *basin cities* hereafter. This feature is referred to as "urban dome" in some previous studies (Idso et al., 1998), albeit we do not adopt this terminology due to the potential confusion of the actual accumulation of $CO_2$ by the concept of a virtual dome of

$CO_2$ over the city. Finally, the results for the Pearl River Delta (PRD) region, China are analyzed to investigate the relative contribution of biogenic fluxes on $X_{CO2}$ enhancements, in addition to increased complexity by the coastal atmospheric dynamics. The PRD metropolitan region is an agglomeration of several cities including Guangzhou, Hong Kong, Shenzhen,



Zhuhai, Dongguan, and Zhongshan etc., which is one of the largest metropolitan regions in the world with about 45 million people and referred to as the PRD region hereafter.

In this paper, the utility of OCO-2 $X_{CO2}$ data to constrain $CO_{2ff}$ emissions from Riyadh, Cairo, Los Angeles and the PRD region are presented. The enhancements in $X_{CO2}$ attributable to urban $CO_{2ff}$ emissions are examined based on OCO-2 $X_{CO2}$
retrievals, and are compared to that obtained from high-resolution modeling by using the Weather Research and Forecasting (WRF) model in chemistry mode (WRF-Chem) for passive tracers, with the $CO_{2ff}$ emissions imposed by the Open-source Data Inventory for Anthropogenic $CO_2$ (ODIAC) (Oda et al., 2017b; Oda and Maksyutov, 2011, 2015). Since the atmospheric transport model errors are one of the major factors influencing inverse modeling, we designed pseudo-data experiments to evaluate the amount of satellite observations needed to constrain $CO_{2ff}$ emissions in presence of random transport model errors
based on high-resolution forward modeling of urban $CO_{2ff}$ for different meteorological conditions. The contributions of biogenic fluxes are assessed for the PRD region by using forward simulations coupled to Net Ecosystem Exchange (NEE) from the Multi-scale Synthesis and Terrestrial Model Intercomparison Project (MsTMIP) (Fisher et al., 2016; Huntzinger et al., 2013). The implications of our results on tracking temporal variability of $CO_{2ff}$ emissions on urban scale are discussed.

## 2 Data and Method

### 2.1 OCO-2 $X_{CO2}$ observations

The OCO-2 daily lite files (v7r) from September 2014 to April 2016 are used in this study (data available online at https://CO2.jpl.nasa.gov/). The OCO-2 satellite operates in a sun-synchronous polar orbit at the altitude of about 705 km and crosses the equator nominally at 13:36 LT (Local Time). It provides high-resolution spectroscopic measurements at eight adjacent 2.25 km long footprints within a narrow swath every 0.333 s, with a cross-track resolution of 0.1~1.3 km, which are
used to retrieve the $X_{CO2}$ (Crisp, 2008; Eldering et al., 2017). The $X_{CO2}$ data provided in lite files are retrieved using the Atmospheric $CO_2$ Observations from Space (ACOS) algorithm (O'Dell et al., 2012) and a bias correction is applied (Mandrake et al., 2015).  We select the $X_{CO2}$ data with Quality Flag (QF) of zero, which is a label for data passing the internal quality check (Mandrake et al., 2015).

### 2.2 Atmospheric modeling of $CO_2$

#### 2.2.1 Atmospheric model simulations

The $X_{CO2}$ fields are simulated by high-resolution forward atmospheric transport modeling  using the Weather Research and Forecasting model coupled with chemistry processes (WRF-Chem) (Grell et al., 2005; Skamarock et al., 2008) version 3.6.1, which is slightly modified for tracking passive tracers (Lauvaux et al., 2012). Table 1 presents a summary of the simulations performed in this study. The model grids are configured separately for each city. One-way nested domains with
resolutions of 27, 9, 3, and 1 km are used for Riyadh and Cairo, 36, 12 and 4 km for LA, and 36, 12, 4, and 1.333 km for the PRD region. All the domains are set up with 51 terrain following vertical levels. The 6-hourly NCEP FNL (Final) Operational Global Analysis data on 0.5 °×0.5 ° grids are used for the initial and boundary conditions of meteorological and land surface





fields. Simulations are conducted for every 4 days with an integration time period of 108 hours, including a spin-up of 12 hours starting at 12:00 UTC on initial day. The simulation results are outputted hourly.

### 2.2.2 Fossil fuel $CO_2$ emissions

Enhancements in $X_{CO2}$ induced by $CO_{2ff}$ emissions ($\Delta X_{CO2ff}$) are estimated by tracking a passive tracer imposed by the
fossil fuel emissions from the Open-source Data Inventory for Atmospheric Carbon dioxide (ODIAC) (Oda et al., 2017; Oda and Maksyutov, 2011, 2015). The ODIAC emission data product is a global 1×1 km gridded monthly fossil fuel $CO_2$ emission inventory, developed based on country level fossil fuel $CO_2$ emission estimates, fuel consumption statistics, satellite-observed nightlight data, and point source information (geographical locations and emission intensities) from the CARbon Monitoring for Action (CARMA) power plant database (Oda et al., 2017b). The global nightlight data were used as a geo-referenced,
spatial proxy to determine the spatial extent of anthropogenic emissions from line and diffused (area) sources (e.g. road traffic, residential or commercial fuel consumption). We used the year 2015 version of the ODIAC emission dataset (ODIAC2015a, available at http://db.cger.nies.go.jp/dataset/ODIAC/). The ODIAC gridded emission fields defined on a global rectangular (latitude × longitude) coordinate are remapped to meet the grids resolutions for each simulation domain.

### 2.2.3 Biogenic fluxes in urban areas

Additionally for the PRD region, the contribution of biogenic fluxes on spatial distributions of $X_{CO2}$ ($X_{CO2bio}$) is examined by WRF simulations. Since there is no established urban Net Ecosystem Exchange (NEE) data available, the NEE from 15 different global Terrestrial Biogeochemical Models with spatial resolutions of 0.5°×0.5° in the Multi-scale Synthesis and Terrestrial Model Intercomparison Project (MsTMIP) (Huntzinger et al., 2013) are used to impose the biogenic $CO_2$ fluxes. In order to better characterize the diurnal variability and spatial distribution of biogenic fluxes in and around the cities, a 3-hourly
dataset for global biogenic fluxes (Fisher et al., 2016) is used, which is temporally downscaled from the monthly global models. Furthermore, we spatially downscale the 3-hourly NEE from the original 0.5°×0.5° MsTMIP grids (e.g. Fig. 1a) to the WRF grids using the Green Vegetation Fraction (GVF), with the assumption that vegetation productivity and respiration scales linearly with canopy coverage. A robust relationship between canopy cover and biomass was observed in Boston, which supports the use of GVF as a proxy for biomass, and hence as a scaling parameter for biogenic fluxes (Briber et al., 2013;
Raciti et al., 2012). Using high resolution vegetation greenness, the NEE is downscaled as follows:

$$E_{wrf, i, j} = (E_{blin, i, j}/GVF_{blin, i, j}) \times GVF_{wrf, i, j}$$

where the subscripts $i, j$ represent the coordinates of a WRF grid cell, $E_{wrf}$ the NEE at WRF grids (e.g. Fig. 1c), and $E_{blin}$ the bilinear interpolated NEE from the original 0.5°×0.5° grids to WRF grids (e.g. Fig. 1b). $GVF_{blin}$ is interpolated using MODIS climatological observations of GVF from 2001 to 2010 (e.g. Fig. 1d) in a similar way of deriving $E_{blin}$ (e.g. Fig. 1c), ensuring
the same spatial representativeness of $GVF_{blin}$ and $E_{blin}$, and $GVF_{wrf}$ (e.g. Fig. 1e) is the GVF projected on the WRF grid. The uncertainties in biogenic $CO_2$ are represented by the variability among the simulated $X_{CO2bio}$ concentrations of the 15 members.





2.3 Observing System Simulation Experiments (OSSEs)

In order to assess the potential of OCO-2-like satellites to quantify $CO_{2ff}$ emissions from cities, we implement several Observing System Simulation Experiments (OSSEs) based on forward modeling of meteorological fields and column $X_{CO2}$ ($X_{CO2ff}$ and $X_{CO2bio}$).Here we focus on the errors in the $CO_{2ff}$ emissions estimates resulting from transport model errors with
constraints by $X_{CO2}$ measurements. The prescribed ODIAC fossil fuel $CO_2$ emissions are assumed as the "true fluxes" to be retrieved. The impact of errors in prior emissions is not included here, which is limited by the lack of high-resolution error structure of the emission maps for most cities. For Riyadh and Cairo, which are *plume cities*, the hourly forward modeling results of $X_{CO2}$ in daytime hours (09:00-15:00 LST) with the angle between $X_{CO2}$ plume and ground track ranging between $10°$ and $170°$are used to construct pseudo observations of $X_{CO2}$ by interpolating the model results to a typical ground track in nadir
mode. Here the retrieval errors are not included. The construction of pseudo modeling data of $X_{CO2}$ are detailed in the following section to represent the transport model errors (section 2.3.2). The uncertainty in the retrieved emissions is assessed by performing multiple emission optimizations (Monte-Carlo approach). Assuming *n* available OCO-2 overpasses over a city during a certain time period, we randomly select *n* samples of pseudo observations (i.e. perfect modeled $X_{CO2}$) and modeling data (i.e. perturbed modeled $X_{CO2}$), and calculate the optimal emissions by minimizing the mismatch between the total $X_{CO2}$
enhancements ($\Delta X_{CO2}$) derived from pseudo observations and modeling (see section 2.3.1).

For Los Angeles, representing a typical *basin city*, the OSSEs are conducted using ensemble of WRF simulations with varying PBL and urban canopy physics parameterizations (Feng et al., 2016). The optimization of $CO_{2ff}$ emissions is conducted by estimating the statistical distribution of the emission scaling factor, which is detailed in section 2.3.2. Note that for the OSSEs, background $X_{CO2}$ concentrations are presumed to be well known, which is not represented in the pseudo data. The
enhancements ($\Delta X_{CO2ff}$) only reflect contributions from local fossil fuel $CO_2$ emissions. However, for in-situ observations the determination of background $CO_2$ concentrations is more complicated and crucial to interpret observed signals attributable to $CO_{2ff}$ emissions, which will be discussed in section 3.4 for real-world cases.

2.3.1 Emission optimization

The urban $CO_{2ff}$ emissions are optimized using a simple method by adjusting total emissions from the city with a scaling factor (*S*), which is calculated by scaling the total $X_{CO2ff}$ enhancements ($\Delta X_{CO2ff}$) of the pseudo modeling (integrated along the ground track) to match that of the pseudo observations:

$$S = \sum_{lat} \Delta X_{CO2ff,o} / \sum_{lat} \Delta X_{CO2ff,m}$$

where the subscript *m* represents data from modeling, an d *o* for observation. Since $CO_2$ is a passive tracer, $\Delta X_{CO2ff}$ is expected
to be linearly scaled with emissions. Therefore, the emissions are optimized on a total basis, namely the optimized $CO_{2ff}$ emission from a city is the product of the a priori total emission and the scaling factor. It should be noted that, the optimization method used here is a "displaced optimization", which artificially alleviates the impact of wind direction errors in transport modeling. Meanwhile this method yields to nearly unbiased distribution of *S* associated with unbiased wind direction errors, compared to the biased *S* when using the typical least square error method. The justification of the total $X_{CO2}$ optimization is
detailed in the Supplement.



### 2.3.2 Representation of transport model errors

Transport model errors are represented here using two different approaches: by modifying directly atmospheric $X_{CO2}$ structures from *plume cities*, and by a physics-based ensemble modeling for *basin cities*. Both methods are evaluated using surface meteorological measurements to calibrate the statistics of the model to realistic transport errors.

a. *Plume city*: direct perturbations

For *plume cities*, the enhancements in $X_{CO2}$ related to urban $CO_{2ff}$ emissions ($\Delta X_{CO2ff}$) are dominated by atmospheric transport. In order to propagate errors from simulated wind fields into $\Delta X_{CO2ff}$, the simulated plumes are modified based on wind speed and direction errors, determined following statistics in previous studies (e.g. Deng et al., 2017). Examples for the transport error representation method are presented in Fig. 2 for a simulated $X_{CO2}$ plume over Riyadh. The effects of
positive/negative wind speed errors are represented by stretching (Fig. 2b)/shrinking (Fig. 2c) the $X_{CO2}$ plumes along the average wind direction within the entire domain, and multiplied by a factor $k$ as:

$$k=(U+U_{err})/U$$

where $U$ is the average wind speed, and $U_{err}$ is the randomly specified error of $U$. To incorporate the wind direction errors, plumes are rotated around the emission center of the entire city by an angle of the random error term (Fig. 2d). The errors are
randomly selected from normal distributions of N(0, 1.0) (unit: $ms^{-1}$) for wind speed and N(0, 15) (unit: $\circ$) for wind direction, respectively. This method represents the overall effect of the transport model errors in the lower fraction of the troposphere.

b. *Basin city*: physics-based ensemble

For cities located in basins or valleys, the diffusions of fossil-fuel $CO_2$ are confined by local topography, which violates
the logic of the error propagation method used here for *plume cities*. Therefore, an ensemble approach is used to represent transport model errors, constructed by using multiple combinations of planetary boundary layer (PBL) schemes and urban canopy models (cf. Table 2). The Mellor-Yamada-Nakanishi-Niino (MYNN) 2.5 (Nakanishi and Niino, 2004)(Nakanishi and Niino, 2004) scheme, the Mellor-Yamada-Jancic (MYJ) scheme (Janjić, 1994), and the Bougeault and Lacarrère (BouLac) (Bougeault and Lacarrere, 1989) scheme are used for PBL parameterization. The single-layer urban canopy model (UCM)
(KUSAKA and KIMURA, 2004) and the multi-layer Building Environment Parameterization (BEP) (Martilli et al., 2002) are used for the land surface processes in urban canopy.

In order to ensure that the transport model errors (represented by the ensemble spread) are comparable with the observed model-data mismatches, we evaluated the performance of the overall ensemble by comparing modeling results with surface wind observations. The evaluation of Planetary Boundary Layers Heights (PBLH) was not performed, since the errors in
vertical mixing near the surface does not have much impact on that of total column average concentrations of $X_{CO2}$, though the study by Feng et al. (2016) suggests that PBLH were correctly represented in LA with specific model configurations, but with an overall overestimation of PBLH across the ensemble. Meteorological observations of wind speed and wind direction at 43 synoptic weather stations located within the 4-km domain covering Los Angeles are used for the evaluation. The surface observations are derived from the global hourly Integrated Surface Data (ISD), accessible online at the National Centers for
Environmental Information (NCEI)
([https://gis.ncdc.noaa.gov/geoportal/catalog/search/resource/details.page?id=gov.noaa.ncdc:C00532](https://gis.ncdc.noaa.gov/geoportal/catalog/search/resource/details.page?id=gov.noaa.ncdc:C00532) ).





Since we use displaced optimization based on scaling the total enhancement of $X_{CO2}$ along the observation track (cf. section 2.3.1), the wind direction errors have less impact than wind speed errors as this approach compensates for peak offset along the OCO-2 tracks. We focus here on the wind speed ensemble results to examine the representation of transport model errors for Los Angeles. The model spread is represented by 1) the standard deviation (STD) of the ensemble, and 2) by half of

the full range of the model results (half difference between the maximum and minimum), as shown in Fig. 3. The original six members exhibit lower spread of wind speeds compared to the mean absolute error (MAE), suggesting an underestimation of the transport errors (Figs. 3a and 3b). In order to enlarge the ensemble spread, we included model results one hour before and after (±l h) the observation time by taking temporal offset into consideration; therefore, the ensemble size is increased to 18. The ensemble spreads for wind speed and wind direction both match the observed model-data differences (MAE) better when

using half of the full ensemble range compared to the STD (Figs. 3c and 3d). Based on these results, the 1-hour lag and the min-max of the ensemble will be used to represent the transport model uncertainties in the pseudo data experiments for LA. The hourly ensemble average distribution of $X_{CO2}$ is interpolated along a typical track in nadir mode and used as the pseudo observations. Assuming that the scaling factor $S$ follows normal distribution of $N(\mu_s, \sigma_s)$, we estimate $S$ by randomly selecting a value according to its distribution. The mean error $\mu_s$ is estimated as the relative bias of the ensemble mean wind speed

compared to observations at the 43 stations. The $\sigma_s$ is estimated as half of the difference between the scaling factors calculated using the ensemble members with the maximum and minimum wind speed as pseudo modeling data.

**3 Results**

3.1 Simulations for plume and basin cities and comparison with OCO-2 data

a. Riyadh and Cairo: *plume cities*

The simulated $X_{CO2ff}$ enhancements ($\Delta X_{CO2ff}$) for Riyadh are characterized by elongated plume structures mainly dependent on atmospheric transport conditions (i.e. wind speed and direction). Figure 4 shows the modeling results of $\Delta X_{CO2ff}$ in the 1-km resolution domain at 09:00, 10:00, and 11:00 UTC December 29, 2014. 10:00 UTC is the approximately overpassing time of OCO-2. We note here that a constant value of $X_{CO2}$ is used as the background $CO_2$ concentration, defined by the observations located outside the city plume. Here the constant value for background $CO_2$ is determined by manually

examining the observations. The simulated $\Delta X_{CO2ff}$ demonstrates rapid-changing and fine-scale structures of the plumes, with discernable variations among the distributions of $\Delta X_{CO2ff}$ within the several hours shown in Fig. 4. The $\Delta X_{CO2ff}$ along the corresponding OCO-2 overpass at 10:00 UTC and one hour before or after (Fig. 5) also suggests the distinctive temporal variations of the plume. Comparing the simulated $\Delta X_{CO2ff}$ from the 1-km, 3-km and 9-km resolution domains, the peak values are lower and smoother when using coarser resolutions, indicating the necessity of using high-resolution simulations to

reproduce structures in the plumes realistically. The modeling results of the 1-km resolution domain are used in our pseudo data experiments with the results presented in section 3.2.

b. Los Angeles: *basin city*

By contrast, over a *basin city* like LA the $X_{CO2}$ enhancements due to fossil fuel emissions are not only affected by the atmospheric dynamics and local emissions, but also by local topography. As an example, Figures 6 and 7 show the simulated



and observed $\Delta X_{CO2ff}$ by the six ensemble members at 21:00 UTC July 6, 2015 in the model domain and along the satellite track. Compared to the observations, the peak is displaced to the north by about 0.4 ° in latitude, which could be related to the bias in the simulated wind field with a stronger northern component. The simulations show diffusive distributions of $\Delta X_{CO2ff}$ over the basin with the peak values located over the eastern end of the city, downwind of the major emissions (Fig. 6). A recent study by (Hedelius et al., 2017) demonstrates persistent differences (~0.8 ppm) in $X_{CO2}$ between two locations only 9 km apart within the LA basin, which are partly explained by the steep terrain responsible for 20–50% of the variability in $X_{CO2}$. This also suggests that the transport model errors could not be represented by rescaling the $\Delta X_{CO2ff}$ over the basin. Therefore, the spread of ensemble modeling with different physical configurations are used to represent the impact of the transport model errors on emission optimization.

3.2 Uncertainty in fossil fuel $CO_2$ emissions: impact of transport model errors and sampling density

   a. Riyadh and Cairo: *plume cities*

   Based on hourly simulated $X_{CO2}$ mole fractions over Riyadh and Cairo, OSSEs are conducted by sampling model results over several weeks from both cities, providing 31 and 20 days (217 and 140 daytime hours) of simulated $X_{CO2}$ under different meteorological conditions. The simulated $X_{CO2}$ plumes are perturbed according to wind field errors (cf. section 2.3.2). The prior total emissions are optimized using the single scaling factor ($S$) in a non-Gaussian optimization system. The optimized total emission is the product of the scaling factor and the total of the priors. The statistical distribution of $S$ is retrieved by randomly selecting our pseudo observation and modeling tracks among the available daytime results. These pseudo data samples are generated for 100 times (i.e. with 100 different perturbations) to provide a complete description of the inverse emissions. Figure 8 shows the relation between the inverse emissions (here represented by the distribution of $S$) starting with a single track up to 30. Here we assume that the true emissions remain constant. The impact of seasonal and diurnal cycles in emissions is discussed in section 5. The average and uncertainty ($\pm 1\sigma$) of $S$ is shown for Riyadh and Cairo in Figs. 8(a) and 8(b). As the number of tracks increases, the average of S approaches 1—i.e., the "true" $S$. In the experiments for *plume cities*, the emission estimates are unbiased, consistent with the nearly unbiased wind speed and wind direction errors shown in previous studies. The uncertainty of $S$ converges rapidly with the increasing amount of available OCO-2 data, indicating the effective constraint on the total emissions by the observations. We note that the pseudo "tracks" used here denote overpasses with $X_{CO2}$ enhancements attributable to local $CO_{2ff}$ emissions. Based on these experiments, about 9~10 tracks are necessary to constrain the uncertainty ($\pm 1\sigma$) in total emissions estimates to reach less than 15%. This suggests that the uncertainties in emission estimates are determined by the amount of observations when the available amount of observations is small (less than 10 tracks). However, with the number of available tracks increasing, the distribution of $S$ reaches a steady condition with $1\sigma$ of approximately 0.07, i.e. an uncertainty of 7% in the emission estimates. Therefore the transport model errors define the lower level of emissions uncertainty of the convergence of $S$, i.e. the improvement of transport model is required in order to further reduce the uncertainty in the total emission estimates.

   b. Los Angeles: *basin city*

   The statistical distribution of the total emissions' scaling factor is examined by exploring the hourly ensemble simulation results of $X_{CO2}$ for the Los Angeles basin, sampling ensemble members as described in section 3.1. The average scaling factor



$S$ shows a positive bias of about 0.133—i.e., an overestimation of about 13% of total emissions compared to the true value ($S$=1). This is associated with the positive bias of surface wind speed for LA, which is a concern for air quality applications reported by previous studies (Angevine et al., 2012; Feng et al., 2016)(Feng et al., 2016; Angevine et al., 2012). The transport model errors represented by the ensemble members for LA lead to less variability in the total enhancements, thus less uncertainty in the scaling factor (cf. Figure 8(c)), compared to that for Riyadh and Cairo. The scaling factor can be constrained at 0.05 uncertainty (or 5%) with 10 tracks. The bias in emissions estimates remains independent of the number of tracks, which is affected by systematic transport model errors.

### 3.3 Uncertainty in fossil fuel $CO_2$ emissions: impact of biogenic fluxes

Forward simulations are conducted for the PRD region to demonstrate the influence of biogenic fluxes and the associated uncertainties on the estimation of total emissions in this region. As described in section 2.2.3, the biogenic fluxes are imposed by 15 MsTMIP flux maps coupled to our WRF modeling system. Following the spatial distribution of $CO_{2ff}$ emissions from several cities located in the PRD region, the modeled $X_{CO2ff}$ enhancements are characterized with multiple *plume cities* features, with long bands of $CO_2$ stretching downwind from the major sources when the atmospheric transport is strong (i.e. high wind speeds) and persistent (i.e. steady winds) as shown in Fig. 9. Compared to the OCO-2 observations (Figs. 9c and 9d), the $X_{CO2}$ enhancements are underestimated by the model. Both tracks show larger variations in $X_{CO2}$. On both days, the coastal circulation contrasts with the continental wind regimes, with fast ocean winds on January 15th and near-zero wind speed on August 4th, opposite to the inland circulation patterns. Since cities in the PRD region are less vegetated compared to the surrounding area (Fig. 1f), the distribution of biogenic fluxes is expected to impact positively the city $X_{CO2}$ enhancements during daytime with stronger photosynthesis surrounding the cities induced by the vegetation gradient, which is demonstrated in a schematic diagram (Fig. 10). This impact is also validated by our simulation results for the two cases (Fig. 9), with the biogenic signals exhibiting enlargements when added to the $\Delta X_{CO2}$ signals. According to the simulated enhancements from the tracers imposed by the 15 biogenic MsTMIP fluxes, the biogenic contributions to the total $X_{CO2}$ enhancement along the OCO-2 track are about $24\pm21\%$ ($1\sigma$) and $19\pm15\%$ ($1\sigma$) for the two cases examined here. This result suggests that if the total enhancement is used to constrain local emissions, and the biogenic contribution is not subtracted, the total emissions would be overestimated by about $32\pm27\%$ and $23\pm18\%$ on these two days. The measurements of $^{14}CO_2$ at the monitoring sites in Los Angeles indicated ~25% of biogenic contributions to the mid-day $CO_2$ enhancement over background (Miller et al., 2017). These results indicate significant influence of biospheric fluxes on local $X_{CO2}$ variability for vegetated areas.

### 3.4 $CO_{2ff}$ emissions estimates using OCO-2 observations

#### a. Riyadh and Cairo: *plume cities*

In order to examine the representation of transport model errors for the emission optimization in real cases, the scaling factor is calculated using OCO-2 $X_{CO2}$ along several tracks in nadir or glint mode collected during the simulated time period for Riyadh and Cairo. The tracks are selected by comparing the observed $X_{CO2}$ with the simulated plumes to ensure that the satellite overpassed downwind of the cities and captured $X_{CO2}$ signals attributable to the local emitters. The transport model errors are incorporated using the same method as for the OSSEs (cf. Section 2.3.2), with sampling size of $10^3$ for each track.





The quantification of emissions requires careful determination of background $X_{CO2}$, which technically represent the atmospheric $X_{CO2}$ abundance without the impact of local fossil-fuel emissions. The $X_{CO2}$ background shows spatial and temporal variations due to long- and medium-range transport of $CO_2$ and would impair our ability to extract enhancements due to $CO_{2ff}$ emissions form observations. Moreover, the contamination of space-based observations due to clouds and aerosols

lead to more difficulties, when the background values on the edges of the urban plume being obscured or deteriorated due to poor data quality. For Riyadh and Cairo linear background $X_{CO2}$ is used. The latitudinal trend of the background line is defined by linear regression of the interested portion of $X_{CO2}$ observations, i.e. within the simulation domain, and the intercept is determined by forcing the background line passing the minimum $X_{CO2}$ among the observations. Figure 11 shows the histograms of scaling factors for the selected tracks over Riyadh and Cairo. The distributions of *S* for Riyadh are non-Gaussian, centered

on higher scaling factors (here 1.6-1.9) suggesting larger emissions than ODIAC emissions imposed. Both Riyadh and Cairo show bimodal distributions for three tracks (27 December, 18 March, and 15 July) due to negative perturbed wind speeds caused by large errors added on small absolute wind speed. The track on December 29 shows also two modes corresponding to two Best Likelihoods from modeled $X_{CO2}$ mole fractions. For the three tracks over Cairo on 28 February, 19 May, and 16 August 2015, the distributions of *S* are centered around 2.4 to 2.9 and consistently over one in all cases, suggesting higher

emissions from the Cairo area.

### b. Los Angeles: *basin city*

Six OCO-2 tracks were examined over LA on July 6 and 15, August 7 and 16, October 10 and 12, 2015. In order to extract fossil-fuel $X_{CO2}$ signals, constant and simulated background $X_{CO2}$ are used. The constant background concentration is derived as the average $X_{CO2}$ of the measurements over the desert north to LA. Meanwhile, the variations of background $X_{CO2}$ were

calculated based on the WRF simulations coupled with the Vegetation Photosynthesis and Respiration Model (VPRM) (Mahadevan et al., 2008) (WRF-VPRM). Figure 12 shows the OCO-2 measurements and the background $X_{CO2}$. The WRF-VPRM background is characterized with nearly linear variation along the tracks. Generally, the two definitions give similar information of background $X_{CO2}$. The WRF-VPRM simulations exhibit better representation of the spatial variations of background $X_{CO2}$ for the tracks on July 15, August 7 and October 10, 2015.

Given the two different definitions of background $X_{CO2}$ shown above, we constrain the total fossil-fuel $CO_2$ emissions from LA using the observations from the six OCO-2 tracks during the modeling time period. Note that the observations along the tracks shown here are chopped near the northern edge of the desert, because the measurements collected north to the desert are usually deteriorated due to the mountainous topography. The scaling factors are calculated as the ratio between the observed total $\Delta X_{CO2}$ and the simulated total $\Delta X_{CO2ff}$, as shown in Figure 14. The uncertainty is estimated as 1σ of the scaling

factors calculated with simulated total $\Delta X_{CO2ff}$ by the 18 ensemble members. Since the monthly emissions are used in the experiments as a priori emissions, we adjust the simulated total $\Delta X_{CO2ff}$ to daytime value at approximately the overpassing time of OCO-2 by scaling with a factor of 1.288, which characterize the diurnal variations of the emissions for ODIAC by (Nassar et al., 2013). The total emission within the 4-km domain covering LA are 103.98 (133.93 with the adjustment, same hereafter) $TgCO_2$ $yr^{-1}$ for July, 105.52 (135.91) $TgCO_2$ $yr^{-1}$ for August, and 99.54 (128.21) $TgCO_2$ $yr^{-1}$ for October. The mean

scaling factor ranges from 0.78 to 1.16 for the tracks on July 6, August 7, and October 12, 2015, but varies from 3.04 to 6.47 for the other three tracks, associated with the underestimated and displaced $\Delta X_{CO2ff}$ peaks from the simulation (Figs. 13b, 13d,



13e) and un-reproduced signals over the ocean (Figs. 13b and 13d) for the tracks in glint mode. The average scaling factor for all of the six selected tracks are 3.04±0.17 and 3.01±0.17 with constant background and WRF-VPRM background, respectively. The WRF-VPRM background $X_{CO2}$ yields to slightly smaller scaling factors than that using the constant background. The 1-σ uncertainties are comparable, which represent the impact of transport model errors.

**4 Discussion**

We implemented OSSEs to examine the contribution of satellite data availability and transport model errors to the emissions optimizations. For *plume cities* of Riyadh and Cairo as examples, the transport model errors are represented by reshaping the $X_{CO2}$ plumes following unbiased errors in wind speed and direction. The uncertainty (1σ) of the emission estimates is constrained to less than 15% with at least 9~10 tracks of pseudo observations. These pseudo tracks are selected

with the requirements of measuring downwind of the city, with the angle between plume and ground track ranging between 10 °~170 °, ensuring that the $X_{CO2ff}$ enhancements attributable to local emissions are captured by observations. Based on OCO-2 data from September 2014 to April 2016 over Riyadh, there are seven tracks that meet these requirements, i.e. about one track per 2.86 months. This suggest that, it would take about 2.1~2.4 years to collect 9~10 tracks. For *basin city* with LA as an example, the transport model errors are represented by using an ensemble of WRF-ODIAC simulations with different model

physics. The scaling factor uncertainty related to transport model errors would be constrained to 5% with 10 tracks. However, the positive bias in emissions estimates can not be compensated by increasing the number of tracks, which is related to systematic transport model errors. These calculations suggest possibility to further improve emissions monitoring using a sufficient number of tracks with accurate transport simulations. More focused observing platforms dedicated to additional urban measurements will be needed, such as OCO-3 (https://oco3.jpl.nasa.gov) and future GOSAT missions.

The a priori emissions used in the optimization system (ODAIC) were considered to be perfectly distributed in space, neglecting the potential uncertainties in emission spatial structures. This shortcoming was further corrected by optimizing the total $X_{CO2}$ enhancement instead of a point-to-point model-data optimization. The total enhancement allows us to compensate for any missing sources by adjusting the enhanced fraction of the $X_{CO2}$ signals, while linear regression or any point-to-point minimization algorithm would have introduced low biases in our results. Due to the absence of information on prior emissions

errors, we implemented a direct optimization with a Monte Carlo approach and did not evaluate the impact of errors in the prior emissions. Due to the lack of physical measurements, direct assessment of errors in gridded emission fields is difficult (Andres et al., 2016). Error assessments of gridded emissions are thus often done using an inter-emission inventories difference as a proxy for uncertainty, especially at an aggregated spatial resolution (Hutchins et al., 2017; Oda et al., 2015). For the case for ODIAC, it is often challenging to find an emission inventory with a comparable spatial resolution, although ideally an emission

evaluation should be done using a detailed emission data like (Gurney et al., 2012). Error correlations among these products are high because of the similitudes in the inputs. Future studies on emission comparisons should address the impact of uncertainties in spatial structures or missing point sources in $CO_{2ff}$ emission products, similar to (Oda et al., 2017a) over the state of California.

The ensemble of WRF-ODIAC simulations over Los Angeles revealed positive systematic errors in wind speed, which

lead to over-dilution of $CO_{2ff}$ and positive bias of scaling factors. The positive wind bias over LA is well documented by



previous studies. Angevine et al. (2012) found a mean wind bias of +1.0-2.0 m/s, and Feng et al. (2016) reported slightly larger bias of +1.5-2.5 m/s, using surface stations and radar measurements across the LA basin. Similar errors have been reported in other studies at different scales and locations, typically when the steep terrain is incorrectly represented. The LA basin presents large elevation gradients from the sea surface to Mount Wilson top. Future studies trying to quantify emissions from *basin*

*cities* should address the presence of biases, not even considering vertical mixing errors and horizontal distribution of $X_{CO2}$ gradients (Ware et al., 2016). Recent urban modeling studies (e.g. Deng et al., 2017) showed that simple data assimilation systems can significantly improve modeling performances and decrease systematic errors to negligible levels (<0.5 m/s). In comparison, WRF simulations over *plume cities* located inland with flat terrain showed better model results in terms of wind speed and direction, with wind speed errors of <1 m/s even without data assimilation around Indianapolis, IN (Deng et al.,

2017), or <0.8 m/s around Paris urban area in France (Lac et al., 2013).

Few studies have addressed the impact of human interventions on urban vegetation (Hutyra et al., 2014), which are likely to increase both Gross Primary Productivity and respiration of ecosystems in urban areas compared to their more natural counterparts. Here the impact of biogenic fluxes is represented with the modeling imposed by the MsTMIP NEE from 15 biogeochemical terrestrial models. The simulation results show that, despite the large $CO_{2ff}$ emissions from the Pearl River

Delta, a significant fraction (24±21% and 19±15% for the two cases shown) of the local $X_{CO2}$ enhancement is driven by the local biogenic fluxes. Urban vegetation models at higher resolution without downscaling would help to more objectively quantify the possible impact of biogenic fluxes, especially for tropical and subtropical cities with various spatial gradients of greenness across in urban and suburban area.

Spatial gradients in background $X_{CO2}$ concentrations matter when deriving local emission signals from satellite

observations along tracks. For Los Angeles, we considered two approaches, i.e. constant versus a varying background as simulated by WRF-VPRM. The tracks are chopped at the northern edge of the desert near LA, in order to exclude the data over the complex terrain and therefore with high warn level. Noise in the OCO-2 measurements associated with precision of single retrievals should also be taken into account, which could obscure the background determination. We have not addressed here a general method in background determination for individual OCO-2 tracks.

For the observation data quality, we analyzed each individual track for this study and considered possible artifacts from complex terrain, aerosols, and clouds. Some cases have been discarded as we suspected strong contaminations from aerosols, as confirmed by CALIPSO data but not filtered using the Quality Flags and Warn Levels. For applications at urban scale, one should address the data quality control by examining the observations, while current filters are mostly applicable on global scale.

The potential biases in emissions estimates introduced by incomplete sampling of satellite measurements have not been considered here. The seasonal and diurnal variations of $CO_{2ff}$ emissions might not be fully tracked, since the satellite soundings are available only in daytime with clear-sky conditions. It has been reported that, satellite $X_{CO2}$ retrievals must be assimilated at the time and location of the observations, since the clear-sky sampling bias leads to underestimated mean $CO_2$ (Corbin and Denning, 2006). To examine its impact on $CO_{2ff}$ emission estimates, we used the hourly $CO_{2ff}$ emissions from Hestia-

Indianapolis product at the scale of buildings and street segments (Gurney et al., 2012), which is one of the most accurate and complete emission inventory available and evaluated against in situ tower measurements (Lauvaux et al., 2016). Low cloud





cover days were determined according to surface observations within the city. The average total emission on low cloud cover days are larger than the that of the full year, indicating positive systematic error of 13% by only sampling low cloud cover days. Seasonal bias will vary by location due to the climatological variations of cloud cover (e.g. monsoon, or rainy season). For the diurnal bias, the Hestia product indicates 14% positive bias for Indianapolis by comparing daytime average (09:00-

14:00 LST) emissions to the daily average. The diurnal variation factors proposed by Nassar et al. (2013) are 1.195, 1.127 and 1.288 for Riyadh, Cairo and LA for the emissions at about the satellite overpassing time against the monthly average, suggesting over-estimations by about 19.5%, 12.7% and 28.8% when sampling daytime only. These cycles could be compensated by optimal sampling strategies but only active sensors will be able to sample across clouds and at night. For future missions, the sampling bias might be compensated by more frequent tracks or targeted view modes.

**5 Conclusions**

In this paper, we examined the potential of $X_{CO_2}$ observations from OCO-2 to quantify $CO_{2ff}$ emissions from urban areas by carrying out OSSEs from high resolution WRF-ODIAC simulations over multiple cities. The capability to constrain emission uncertainties in the presence of transport model errors was examined using Monte-Carlo approach. The OSSEs results show convergence of emission uncertainties with increasing amount of observations. The 1-σ uncertainty of emission estimates

is constrained to approximately 15% and 5% with about 10 pseudo tracks for *plume cities* (Riyadh and Cairo) and basin city (Los Angeles metropolitan area), respectively. The systematic positive wind speed biases in model transport for LA lead to overestimations in the scaling factor and therefore in total $CO_{2ff}$ emissions, which require data assimilation to improve high-resolution atmospheric simulations. By comparison, *plume cities* are more promising with current mesoscale modeling systems. Systematic analysis of cities across the globe will help to refine the needs for quantifying urban $CO_{2ff}$ emissions using

satellite observations around the globe by considering the limitations of each region.

The simulations driven by MsTMIP NEE of the cities in China's Pearl River Delta region indicate that, biogenic fluxes are critical for cities located in well vegetated areas, typically in mid-latitudes to the equatorial areas. Gradients in biogenic fluxes in the urban and surrounding areas in the PRD region contribute to about 24±21% (1σ) and 19±15% (1σ) of the total $X_{CO_2}$ enhancements for the two cases examined, which would lead to overestimation of total emissions by about 32±27% and

23±18%. Similar magnitude was reported with $^{14}CO_2$ measurements in Los Angeles, indicating ~25% of the contribution of biospheric sources to the midday $CO_2$ enhancement over background (Miller et al., 2017).

Background mole fractions of $X_{CO_2}$ for urban areas require more consideration. For Los Angeles two definitions are examined, i.e. constant background using average $X_{CO_2}$ over the desert near LA and background determined by WRF-VPRM modeling. The WRF-VPRM background yields to slightly smaller scaling factors than that using the constant background for

half of the six tracks shown here. More sophisticated biospheric modeling can help to develop a better determination of the background $X_{CO2}$.





**Acknowledgments**

This work has been funded by the National Aeronautics and Space Administration (NASA) Orbiting Carbon Observatory 2 (OCO-2) Science Team (award NNX15AI42G), the National Institute for Standards and Technology (NIST) Indianapolis Flux Experiment (INFLUX) project (award 70NANB10H245), and the National Oceanic and Atmospheric Administration (NOAA) project (grant NA13OAR4310076).

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



Table 1. Summary of WRF-Chem simulations performed in this study

| Objective | City/ Metropolitan region | Grids resolution (km) | Innermost domain size (number of grids) | Simulated time period |
|---|---|---|---|---|
| Plume city | Riyadh | 27, 9, 3, 1 | 201×201 | 1-16 November 2014<br>17 December 2014 – 30 January 2015 |
| | Cairo | 27, 9, 3, 1 | 201×201 | 4-7 October 2014<br>16-19 March 2015<br>17-20 May 2015<br>13-16 July 2015<br>14-17 August 2015 |
| Basin city | Los Angeles metropolitan region | 36, 12, 4 | 207×150 | 3 July – 20 August 2015<br>6-19 October 2015 |
| Impact of biogenic fluxes | Pearl River Delta metropolitan region | 36, 12, 4, 1.333 | 240×240 | 12-15 January 2015<br>1-4 August 2015 |

Table 2. WRF configurations of forward simulations for Los Angeles

| Ensemble member | PBL scheme | Surface layer scheme | Urban Canopy model |
|---|---|---|---|
| MYJ | MYJ | Eta similarity (Janjić Eta) | None |
| MYJ_UCM | MYJ | Eta similarity (Janjić Eta) | Noah UCM |
| MYNN | MYNN | Nakanishi and Niino | None |
| MYNN_UCM | MYNN | Nakanishi and Niino | Noah UCM |
| BouLac_BEP | BouLac | Eta similarity (Janjić Eta) | BEP |
| BouLac_UCM | BouLac | Eta similarity (Janjić Eta) | Noah UCM |




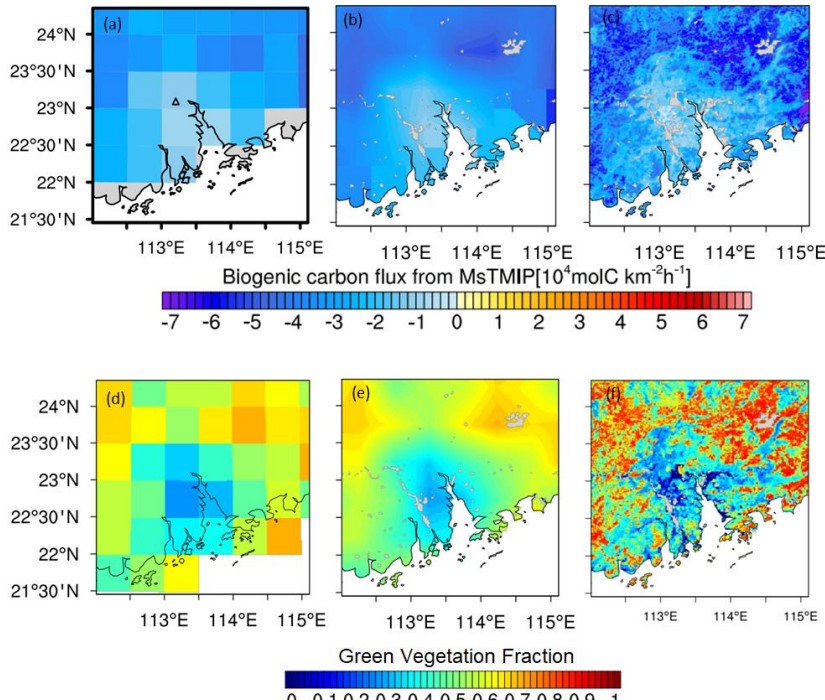

Figure 1. Example of biogenic carbon fluxes (Net Ecosystem Exchange, NEE) downscaling. Top panels show the NEE in PRD region derived from 3-hourly MsTMIP data at 12:00 LT January 12, 2010 on (a) 0.5 °×0.5 ° grid, (b) WRF grid (1.333×1.333 km), derived by bilinear interpolation of original NEE, and (c) WRF grid (1.333×1.333 km), derived by scaling the interpolated NEE. Bottom panels show the green vegetation fraction (GVF) in January on (d) 0.5 °×0.5 ° grid, (e) WRF grid (1×1 km) by bilinear interpolation of GVF in (d), and (f) WRF grid (1×1 km). See texts in section 2.2.3 for further details.



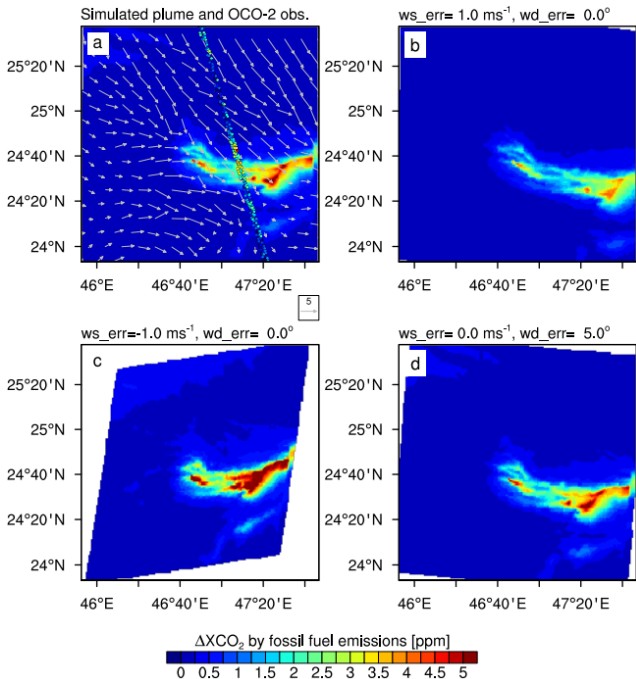

Figure 2. (a) Map of simulated $X_{CO2}$ plume and an OCO-2 ground track in nadir mode at about 10:00 UTC, December 29, 2014; (b) stretched plume to represent a wind speed error of 1.0 ms$^{-1}$; (c) shrunken plume to represent wind speed error of -1.0 m s$^{-1}$; (d) rotated plume to represent wind direction error of 5.0 °.





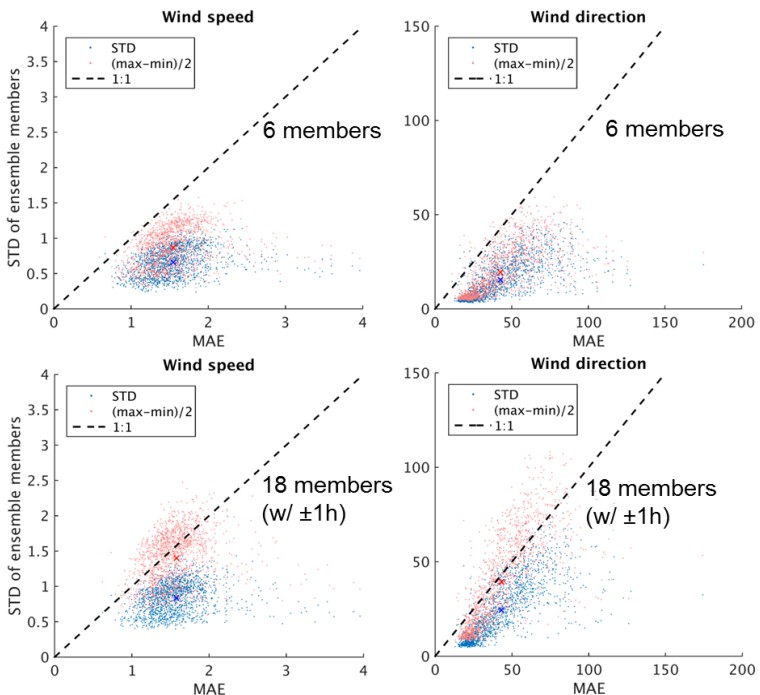

Figure 3. Comparison between the half of ensemble spread and mean absolute error (MAE) of wind speed and wind direction over 43 surface sites in LA and the adjacent area. The half ensemble spread is calculated using the standard deviation (STD, blue scatters) and half of the full range of the model results, i.e. difference between the maximum and minimum values among the ensemble members (red scatters). The top two panels show comparison for the original six members, and the bottom two panels for the 18 members with modeling results at ±1 h included. The red and blue crosses in each panel stand for average points of the scatters in the corresponding color.





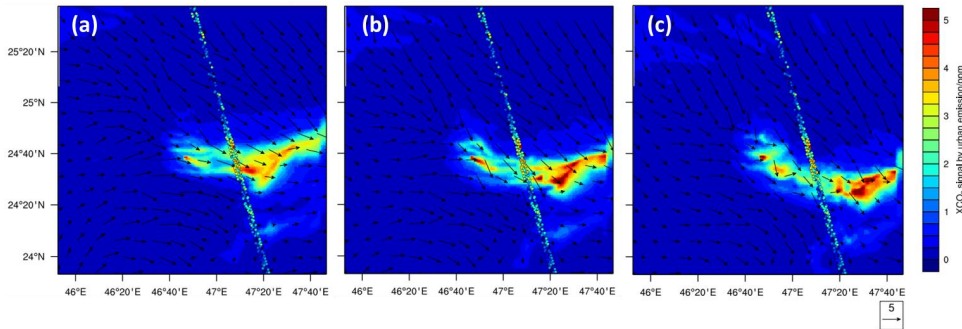

Figure 4. Simulated $\Delta X_{CO2ff}$ of Riyadh and 10-m wind in 1-km resolution domain at (a) 09:00 UTC, (b) 10:00 UTC, and (c) 11:00 UTC December 29, 2014. The colored dots represent the 1-s average OCO-2 data in nadir mode at about 10:00 UTC over this domain, which are filtered with quality flag of zero (QF=0) with a constant value of 397.73 ppm subtracted.

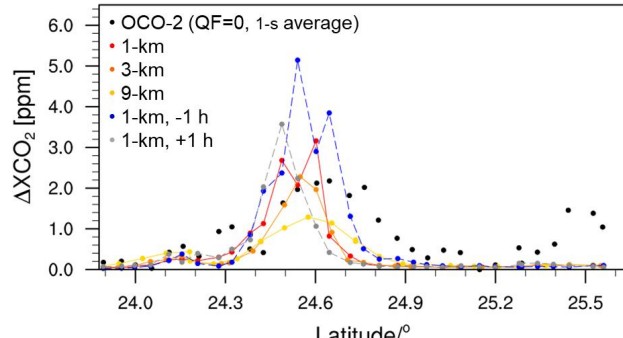

Figure 5. Observed and simulated $\Delta X_{CO2ff}$ along latitude for Riyadh. The black dots represent the OCO-2 data around 10:00 UTC December 29, 2014, filtered with quality flag of zero (QF=0) with a constant value of 397.73 ppm subtracted. The dotted

10 lines in red, orange, and yellow stand for simulations at 10:00 UTC from 1-km, 3-km and 9-km resolution domain. The blue and gray dash-dotted lines represent simulations at ±1 h (09:00 and 10:00 UTC) respectively.



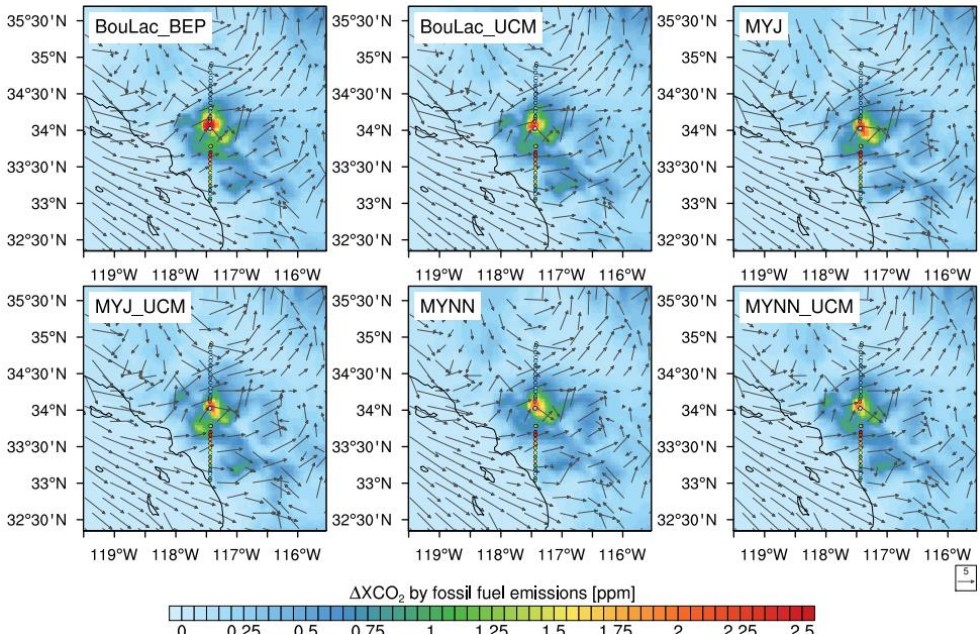

Figure 6. Simulated 10-m wind (arrow vectors) and $\Delta X_{CO_2}$ (color shading) imposed by fossil fuel emissions in LA and in the 4-km resolution domain at 21:00 UTC July 6, 2015. The colored dots represent 1-s average OCO-2 data (nadir mode) collected at about 21:00 UTC, which are filtered by quality flag (QF=0) with a constant value of 400.61 ppm subtracted.

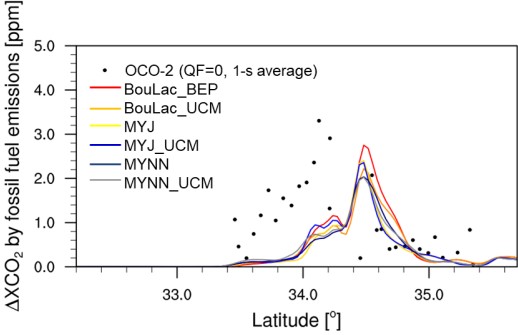

Figure 7. Observed and simulated $\Delta X_{CO_2}$ along latitude for LA. The black dots represent 1-s average OCO-2 data observed at about 21:00 UTC July 6, 2015 (filtered with quality flag of zero (QF=0) with a constant value of 400.61 ppm subtracted). The colored lines stand for simulations of the six ensemble members of the 4-km domain at 21:00 UTC of the same date.



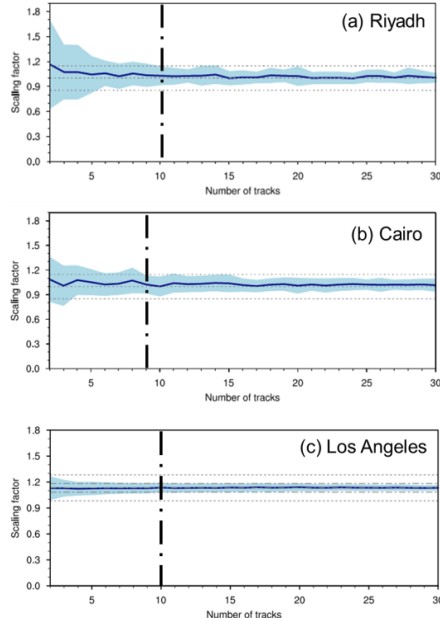

Figure 8. Variation of scaling factor and its uncertainty (±1σ) due to transport errors with increasing number of satellite tracks
for (a) Riyadh, (b) Cairo, and (c) Los Angeles. The dotted lines in each panel represent ±1σ of 0.15, and the dash-dotted lines
in (c) represent ±1σ of 0.05.

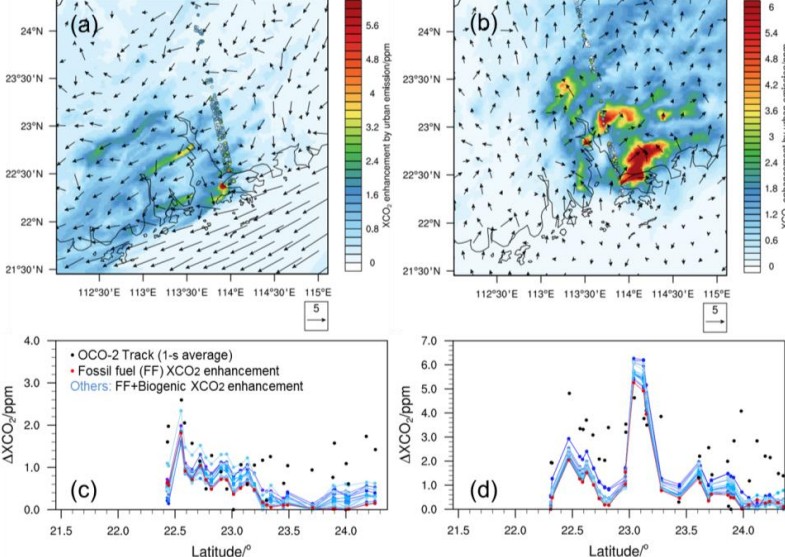

Figure 9. Simulated fossil-fuel $\Delta X_{CO_2}$ due to $CO_{2ff}$ emissions from the PRD region and the 10-m wind vectors in the 1.333-km
resolution domain at (a) 05:00 UTC January 15, 2015, and (b) 05:00 UTC August 4, 2015. The colored dots represent the
OCO-2 data at about 05:00 UTC over this domain, which are filtered with quality flag of zero (QF=0) with a constant value of



399.59 ppm subtracted from data in (a) and 397.74 ppm for (b). The black dots in (c) and (d) show the 1-s average observations of the tracks in (a) and (b), with the modeled fossil-fuel $\Delta X_{CO2}$ enhancements and total $\Delta X_{CO2}$ (owing to both fossil fuel and biogenic fluxes) represented by the red and blue dotted lines, respectively.

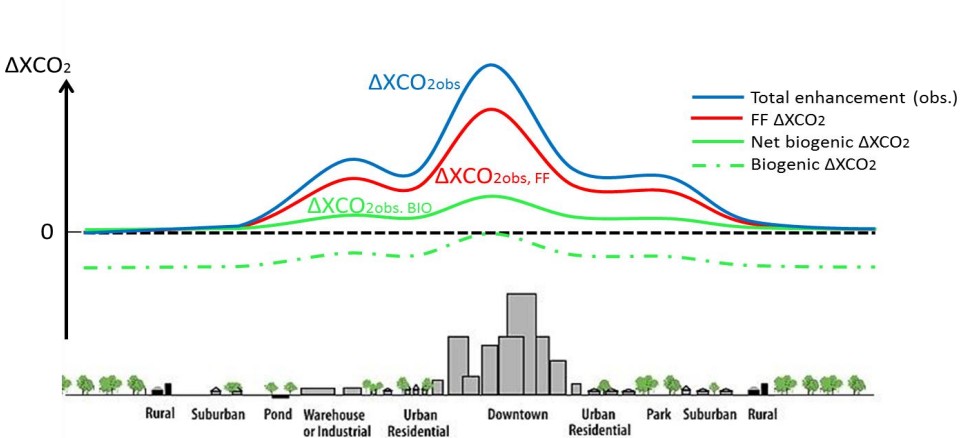

Figure 10. Schematic diagram for the contribution of urban-rural biogenic flux gradient ($\Delta X_{CO2\ obs,\ bio}$) and urban fossil fuel $CO_2$ emissions ($\Delta X_{CO2\ obs,\ ff}$) to the observed $X_{CO2}$ enhancement ($\Delta X_{CO2\ obs}$).

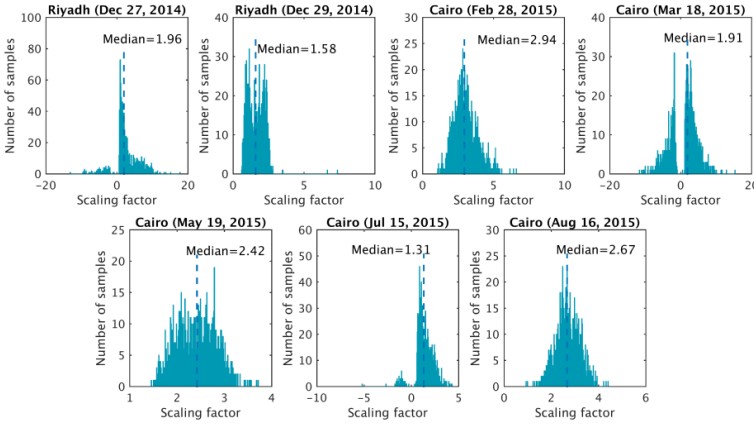

10    Figure 11. Histograms of scaling factor calculated for the tracks on December 27 and 29, 2014 over Riyadh and on February 28, March 18, May 19, July 15, and August 16, 2015 over Cairo. Median values are indicated with the dashed line. The total sample size is $10^3$ for each track.




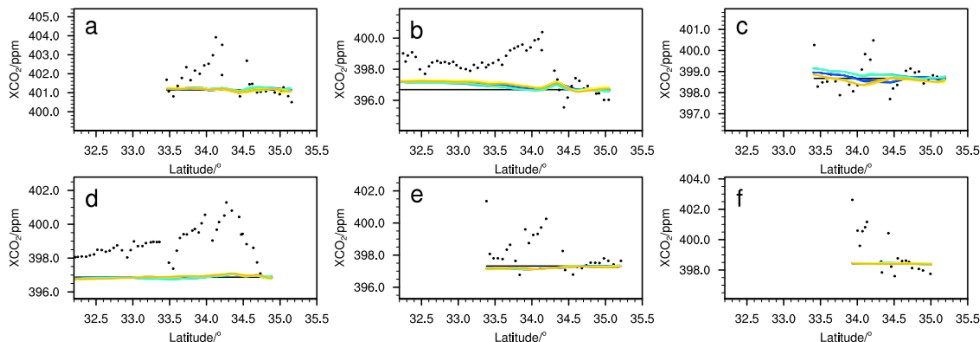

Figure 12. Determination of background $X_{CO2}$ for OCO-2 tracks over LA on (a) July 6, (b) July 15, (c) August 7, (d) August 16, (e) October 10, and (f) October 12, 2015. The black dots stand for 1-s average OCO-2 $X_{CO2}$, filtered with quality flag of zero (QF=0). The black lines represent constant background determined as the average $X_{CO2}$ observed within the desert north to LA, and the colored lines for WRF-VPRM $X_{CO2}$ by the 18 ensemble members, forced to cross the constant background.

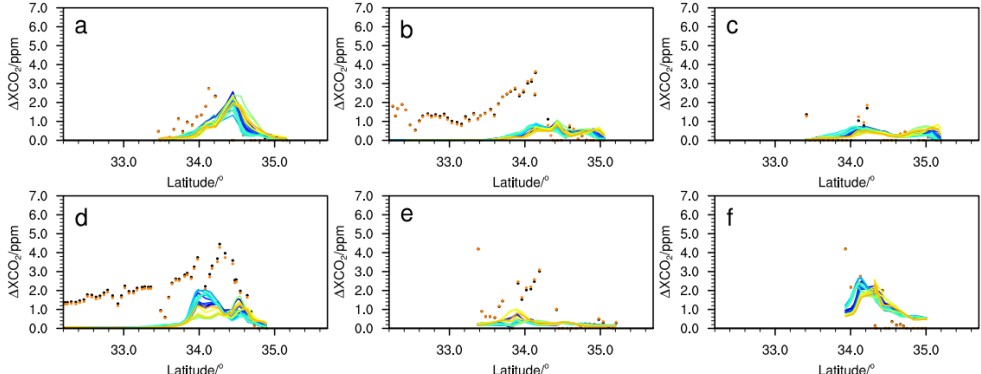

Figure 13. $\Delta X_{CO2ff}$ derived from OCO-2 observations and simulations over LA on (a) July 6, (b) July 15, (c) August 7, (d) August 16, (e) October 10, and (f) October 12, 2015. The black and orange dots represent the $X_{CO2}$ enhancements with background defined by constant and WRF-VPRM modeling, respectively. The colored lines represent the simulated $\Delta X_{CO2ff}$ by the 18 ensemble members.





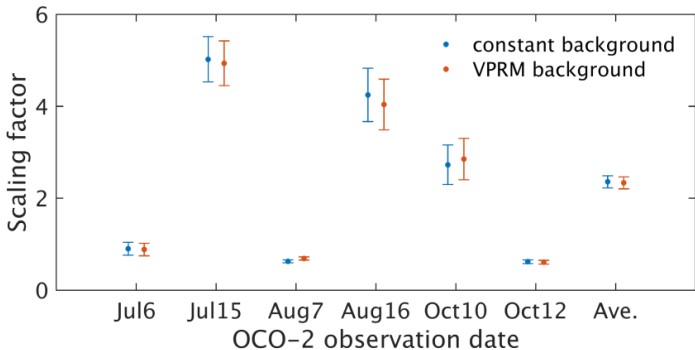

Figure 14. Scaling factor of the total fossil fuel $CO_2$ emissions in LA calculated using $\Delta X_{CO2ff}$ from OCO-2 observations and modeling results. The background $X_{CO2}$ is determined by the average $X_{CO2}$ within the desert north to LA basin (blue) and the WRF-VPRM modeling (red). The dots and the error bars stand for the average and uncertainty ($\pm 1\sigma$) of scaling factors calculated using the 18 ensemble members.