# Peer review of "Constraining fossil fuel CO2 emissions from urban area using OCO-2 observations of total column CO2"

_Atmospheric Chemistry and Physics, 2017_

## Referee Comment (RC1) · Anonymous Referee #2 · 31 Jan 2018

The manuscript "constraining fossil fuel CO2 emissions from urban area using OCO-2 observations of total column CO2" by Ye et al. studies the inversions of urban CO2 emissions based on CO2 data from the OCO-2 mission. It analyzes both Observing System Simulation Experiments and experiments with the real CO2 data from the OCO-2 mission to assess the level of precision and accuracy that could be expected from such inversions.

This is a critical topic in a context of the preparation of various missions for the monitoring of CO2 anthropogenic emissions from space while there is still a lack of studies addressing the feasibility of this monitoring in details. This study gathers tools and ma-

terial that can feed the understanding and discussions in this research field. I think that the authors can produce an important paper with such a basis.

However, I also think that the text needs a strong improvement and that critical issues need to be addressed:

1) Most of the text needs a careful rewriting. At first sight, it looks concise. But when reading it into details, especially when trying to understand how some of the curious results (see below) have been obtained, it appears to be confusing, hasty, approximative, full of gaps, and unstructured. A major issue is the lack of logical flow and of robust focuses. We often do not know where the authors go, and why they make their choices. Large parts of the manuscript look like piles of independent pieces of explanations, analysis and discussions that have not been linked or synthesized. Some of the comments below illustrate this. The second half of the abstract, sections 1 (the introduction), 4 and 5 give an illustration of the lack of logical structure. The second half of the abstract, sections 2.3 and 3.2 (among others) provide a good illustration of the lack of clarity, especially when dealing with the experimental protocol.

2) I do not understand central results and figures of this manuscript. It may be because I do not really understand the details of the computations.

- section 3.2.a and Figure 8: I do not understand what is done when several tracks from different days are used together. Do the authors gather the corresponding 100 perturbed wind conditions together, forming a much bigger ensemble of perturbed wind conditions, and then look at the distribution of S arising from this larger ensemble ? In that case, the convergence they show in Figure 8 would only be an illustration of the decrease of the sampling errors for the distribution of the transport error when increasing the sampling size, i.e., nothing about the asset of using different OCO2 tracks. Or do the authors do things in such a way that N tracks bring N independent statistical transport errors together, which would decrease the overall weight of the transport error ? Then, in the absence of prior and measurement error, the curve on

[Figure]

Figure 8 should converge towards 0. My lack of understanding is a real concern since this is one of the main results that are cited in the discussion, conclusion and abstract.

- section 3.2.b: an OSSE is conducted with synthetic perturbations of the transport. I cannot understand how the bias in the results can be related to a bias of WRF in the real world. My very simple understanding is that this bias is just a consequence of the way the perturbations are chosen and applied in the synthetic world, combined with the non linearity of WRF. Again, this corresponds to one of the main results in the abstract, discussion and conclusion.

- section 3.4 and Figure 11: the text at lines 11-12 page 11 says something really strange for me: "bimodal distributions for three tracks due to negative perturbed wind speeds caused by large errors added on small absolute wind speed". What is a negative perturbed wind speed ? What do the authors do with such a wind speed ? Furthermore, they are a lot of negative S in Figure 11 (the text does not speak about it). How can it be possible ? Would that mean that the authors diagnose negative enhancements of XCO2 along the simulated or observed OCO-2 tracks ? I would not understand it but I do not really know how the authors diagnose the enhancements since section 2.3.1 and the corresponding equation on page 6 lack of rigor. Is there a link between these negative scaling factors (turning cities into sinks of CO2) and the "negative perturbed wind speed" ?

3) The study of the uncertainty in the ecosystem fluxes in the PRD area does not fit with the rest of the paper and the authors did not make much effort to connect the results and discussions corresponding to this study to the other ones. I think that this should be removed, which would help improve the structure and logical flow of the paper. We get a strong feeling that the authors wanted to value an experiment they had done in the PRD region by inserting it artificially into an independent paper. It is all the more problematic that the dissymmetry between the experiments in the "plume cities" and LA is already a bit difficult to admit (with these two cases only, we can already get the feeling of following two independent studies in parallel). There is no justification for not

applying the range of tests on the transport errors to the PRD area. And there is no reason for not applying the tests applied in the PRD area on the biogenic fluxes to, at least, LA, and, maybe, Cairo. Using VPRM in LA when assimilating real data vs. using an ensemble of larger scale products for the OSSEs in the PRD area (i.e., different types of tools) emphasize the feeling that the PRD sections are disconnected from the rest. At last, I think that the analysis in 3.3 are unclear and debatable so it would have to be extended and improved. On a similar topic: the oscillation of the text between distinguishing or mixing the background and biogenic components is problematic, especially since the background is defined several times in different ways in this paper. This, in addition to the problem stated above, leads to strange sentences at the end of the paper ("biogenic fluxes are critical for cities located in well vegetated areas . . . Background mole fractions of XCO2 for urban areas require more consideration . . . More sophisticated biospheric modeling can help to develop a better determination of the background XCO2").

4) The assumption that a change in the wind speed can be simulated by stretching the XCO2 images and rescaling the concentrations needs to be discussed and justified. I think that this assumption is not obvious and that is does not perfectly fit with many model formulations, such as a large range of Gaussian ones.

5) The paper focuses on transport, biogenic and background errors. This justifies the lack of discussion on the measurement errors in the sections dedicated to the OSSEs. However, these measurement errors are nearly ignored when analyzing the results with real data, in the discussion, in the conclusion and even in the introduction (e.g. on p3), while other studies indicate that this is a critical component of the problem. As a consequence, the presentation of the conclusions from section 3 often seems biased and misleading. E.g., lines 11-18 p14 (but also lines 18-19 p1 in the abstract) do not really say that the 15%/5% numbers relate to transport errors only and that the potential of the OCO2 data themselves (i.e. not that of the modeling framework) is not fully investigated. This could strongly favors over-optimistic reading and citations

of this paper (such as when this paper itself summarizes in a very optimistic way the conclusions from Hakkarainen et al. 2016 on p3 line 3). In such a context, I also think that the title of the manuscript itself is misleading.

6) Section 4 does not deliver the type of discussions that are expected when reaching the end of section 3. Section 3 does not go really deep into the result analysis, in particular when looking at the tests with real data (while these tests will certainly be brought forward when citing this paper). Therefore, section 4 should provide more insights on the results. However, the first paragraph in section 4 is merely (if we except 2,3 lines) a summary of what has already been said in section 3, and the rest of section 4 is the piling of small and independent discussions on the perspectives.

7) I could list a lot of minor issues. However, nearly all of them relate to comment 1 and I will thus wait for a later step before undertaking a detailed list. That said, I would like to mention:

- that the references could be complemented by Schwander et al. 2017 (Schwandner, F. M. et al., Spaceborne detection of localized carbon dioxide sources, Science, 358 eaarn5782, 2017) which could potentially impact lines 4-5 p3. Nassar et al. 2017 (Nassar, R., et al., Quantifying CO2 Emissions From Individual Power Plants From Space, Geophysical Research Letters, 44, 10045-10053, 10.1002/2017GL074702, 2017) is also relevant for the discussion on page 3 since power plants and cities are sometimes considered as similar targets and since the quantification of their emissions using OCO2 data raise similar challenges.

- section 2.2.1 and table 1: the information about the spatial resolution of the outer domains is not really interesting if there is no information about their spatial extent.

- regarding the integration of the data along the OCO2 tracks: as said before, the section 2.3.1 and its equation are unclear; I did not understand what is the spatial and/or temporal representativity of the dots in figure 5; and I am not sure about the meaning of the spatial location of the OCO2 dots in figure 6.

- The paper often considers the temporal variability of the emissions as a side problem while one of the central tests studies the asset of using several satellite tracks. The day to day variability of the emissions is merely ignored. The seasonal variations are mentioned but they are not confronted to the statements regarding the number of OCO2 tracks required to balance the transport errors, nor to the actual number of available tracks per year.

- p7 : lines 29-31: i) due to the variations of the wind as a function of the vertical (including wind curl in the PBL), modifying the PBL should have an impact on the XCO2 fields ii) if the PBLH and vertical mixing near the surface do not have much impact, why having perturbed the PBL schemes in these experiments ?

- p11 l 32 If I am not wrong, such a scaling was not applied for Cairo and Riyadh: why ? how to rely in this factor 1.288 while the inventory are assumed to be uncertain ? why not just acknowledging that the estimations correspond to the satellite track times and discussing the extrapolation into daily budgets later ?

- section 3: I feel that the results when using real data (especially in LA) raise severe doubts regarding the potential of the inversion strategy or of the current OCO-2 data, and that this is not highlighted in the conclusion and abstract of the paper. At least, this deserved more discussions in section 4.

- there are a lot of awkward sentences throughout the text such as "These cycles could be compensated by optimal sampling strategies but only active sensors will be able to sample across clouds and at night. For future missions, the sampling bias might be compensated by more frequent tracks or targeted view modes." at the end of section 4
* * *

---

## Referee Comment (RC2) · Anonymous Referee #3 · 1 Mar 2018

This study investigates the potential of observations of total column CO2 (XCO2) from a satellite of type OCO-2 to quantify CO2 emissions from individual cities. The study is based on high -resolution (kilometer scale) urban plume simulations with WRF-CHEM used to conduct observation system simulation experiments and for comparison with real OCO-2 observations.

The topic is highly relevant and timely as several new CO2 satellite missions are currently being planned and more and more applications of OCO-2 data are being publishes. The manuscript is thus a welcome contribution.

The manuscript is fairly well written and the methods, centered around comprehen-

sive model simulations, are generally sound. The study presents many interesting and innovative aspects that clearly deserve being published. In particular, the study investigates uncertainties in the emission estimates related to transport uncertainties by stretching/squeezing or rotating the simulated model fields, applies an ensemble approach for basin cities, and investigates the influence of biospheric fluxes using an ensemble of biosphere flux models downscaled to high resolution for the simulations. The simulation setup is impressive and the analyses presented are comprehensive.

Despite these positive aspects, I also have a few concerns primarily related to the way the study is presented and the conclusions that are drawn as detailed in the following.

Main points:

Based on the abstract, a reader may conclude that OCO-2 is a satellite highly suitable for quantifying CO2 emissions from cities , but OCO-2 has not been designed for this purpose and is clearly far from ideal. The main problem of the manuscript is that the severe limitations of OCO-2 in terms of temporal and spatial coverage are not clearly discussed and that, therefore, a much too optimistic picture is drawn of what can be achieved with such a satellite. The only sentence in the introduction addressing the issue of coverage is the following fairly neutral statement: "discernible CO2 emission imprints can be limited due to the contamination by clouds and aerosols and limitations of spatial-temporal sampling coverage for local sources related to the revisit cycle of sun-synchronous polar orbit and the narrow tracks". OCO-2 has about 15 orbits per day, each with a swath of approx. 10 km. In 1 day it thus covers a total east-west extent of 150 km. For comparison, the circumference of the Earth is about 40'000 km, i.e. OCO- 2 would take 266 days to sample each point on the globe (at the equator) at least once. But OCO-2 has a 16 days repeat cycle which means that many points on the globe will never be observed at all. To measure the plume of a city does not necessarily require flying directly over the city (except for basin cities), but the overpass should be close to have sufficient signal and to be able to unambiguously attribute the plume to its source. On page 2, L35, the authors state that "OCO-2 pioneered the

contiguous high-resolution mapping of global CO2 concentrations", which needs to be changed. A swath of 8 pixels across-track definitely does not qualify OCO-2 as a "mapping mission". OCO-2 takes high-resolution measurements along a narrow line. Its sampling strategy is much closer to a 1D than a 2D mission, which is also why this study analyzes CO2 along individual (1D) lines and not in (2D) images of entire plumes. It is important to make this distinction because several true imaging missions are currently being planned including geostationary and polar orbiting missions.

There are also other reasons why the overall tone of the manuscript is much too optimistic: On page 3 the manuscript states that "a handful of cities with different typical XCO2 features in XCO2ff enhancements" were selected, which gives the impression that the selection was more or less arbitrary and that any other combinations of cities might have worked equally well. However, the authors have picked highly ideal cities with a) very large emissions, b) little interference with biospheric fluxes (Riyadh and Cairo), c) very low average cloud cover, and d) nicely isolated from other cities avoiding overlapping plumes. For the study of interferences with the biosphere, the Pearl River Delta region has been selected, one of the most densely populated regions where, again, anthropogenic emissions are unusually large compared to biospheric influences. There are good reasons for selecting these cities because OCO-2 offers many opportunities to observe their plumes, but it has to be openly communicated why these were selected and that the challenges for most other cities (probably for 99% of all cities of the globe) will be much larger due to frequent cloud coverage, strong interferences with biospheric fluxes (especially during summer coinciding with periods of low cloud cover and hence good observation opportunities), poor coverage due to the OCO-2 orbit geometry and narrow swath, overlapping plumes, plumes below the detection limit etc. Based on the OSSEs it is concluded that emission uncertainties are constrained to less than 15% with at least 9-10 tracks for plume cities and even down to 5% for a basin city, and that it would only take about 2.1-2.4 years to collect a sufficient number of tracks with an OCO-2 type instrument. However, this conclusion is only valid for the unrealistic case of negligible observation uncertainties , perfectly

known background and emission distributions, and non-existence of clouds. In reality, the magnitude of the plumes will typically be in the low ppm to sub-ppm range and hence in a similar range as instrument noise, and clouds will frequently obscure the view. Fitting real observations to the simulations is thus much more difficult and many more overpasses will be needed to reach such a low uncertainty. This is also nicely demonstrated by the real cases presented in Section 3.4, where three overpasses over Riyadh provide median emission scaling factors differing by almost a factor of two (between 1.58 and 2.94), and the histograms even include negative emissions. These real OCO-2 observation cases would offer a nice opportunity to place the previous theoretical OSSE analyses in context and to explain the additional challenges, but there is no discussion of this at all.

Thus, my general recommendations are

a) better emphasize the challenges as well as the limitations of OCO-2

b) stress clearly in the abstract and conclusions that the OSSEs were conducted under idealized conditions neglecting instrument uncertainties and that the convergence to XX% with YY tracks only refers to the contribution of transport uncertainties under these conditions

c) explain that the cities selected in this study were chosen for their ideal properties to demonstrate the potential of OCO-2, but at the same time point out that for many other cities it will remain a challenge to quantify urban emissions with sufficient accuracy.

d) place the results obtained for the real cases in context with the OSSEs

Finally, I would like to point out that Figure 10 can not be published in its current form, since the lower part of the figure has been borrowed from another publication most likely violating copyrights.

Small issues:

Page 1, Line 21: Change "in urban and rural area of Pearl River Delta" -> "in the urban

and rural area of the Pearl River Delta"

P2, L2: Although 70% of global CO2 emissions may be due to energy consumption in cities, a considerable (yet probably unknown) fraction of these emissions do not occur inside the cities but outside in power plants delivering the energy for the cities. This is usually forgotten when just citing this number.

P2, L6: "few exception such as (Gurney et al, 2012)" -> " few exceptions such as Gurney et al. (2012)". There are actually many other locations where the reference is not properly formatted.

P2, L8: "spatial explicit" -> "spatially explicit"

P2, L15: "Inverse modelling, or top-down approach assimilate" -> "Inverse modelling, often referred to as top-down approach, assimilates"

P2, L19: "by inversion method" -> "by inversion methods"

P2, L26: "are detected" -> "have been detected"

P2, L31: "background area" -> "background areas"

P3, L10: Change to ".. have been identified as major sources of .."

P3, paragraph 2: One more important challenge needs to be added, namely the temporal variation of emissions at diurnal to seasonal time scales, which can not be fully captured by a satellite leading to potential biases in the emission estimates (both to diurnal and seasonal sampling biases).

P3, L33: "is referred to" -> "was referred to"

Section 2.2.2: It needs to be mentioned here that no temporal variability of emissions was considered. Furthermore, it should also be mentioned that all CO2 was released at the surface or, if not, how emissions were distributed vertically.

P6, L16: "using ensemble" -> "using an ensemble"

P6, line 29: "an d o" -> "and o"

P6, line 32: What do you mean by "artificially"? This doesn't seem to be the right word here.

P7, L19: "the diffusions of fossil-fuel CO2 are" ->" the diffusion of fossil fuel CO2 is"

P7, L26: "in urban canopy" -> "in the urban canopy"

P7, L33: "Los Angeles are used" -> "Los Angeles were used"

P7, L34: "observations are derived" -> "observations were derived"

P8, L22: "is the approximately overpassing time" -> "is the approximate overpassing time"

P9, L33: "with a stronger northern" -> "with a too strong northern" (sounds better to me)

P10, L3: references appear twice

P10, L12: "characterized with" -> "characterized by"

P11, L6: "and Cairo linear" -> "and Cairo a linear"

P12, L13: "this suggest" -> "this suggests"

P12, L19: Please add CarbonSat (Buchwitz, M., M. Reuter, H. Bovensmann, D. Pillai, J. Heymann, O. Schneising, V. Rozanov, T. Krings, J. P. Burrows, H. Boesch, C. Gerbig, Y. Meijer, and A. Loescher, Carbon Monitoring Satellite (CarbonSat): assessment of atmospheric CO2 and CH4 retrieval errors by error parameterization, Atmos. Meas. Tech., 6, 3477-3500, 2013).

P13, L18: "across in urban" -> "across the urban"

P14, L13: "using Monte Carlo" ->"using a Monte Carlo"

P14, L25: "Similar magnitude" -> "A similar magnitude"

---

## Author Comment (AC1) · 5 Jun 2018

**Responses to interactive comments on "Constraining fossil fuel $CO_2$ emissions from urban area using OCO-2 observations of total column $CO_2$" by Ye et al.**

We thank the referees for reviewing the manuscript and for their dedication to help improve this manuscript. Their comments are copied below along with our responses (in blue).

**RC1 by Anonymous Referee #2**

The manuscript "constraining fossil fuel $CO_2$ emissions from urban area using OCO-2 observations of total column $CO_2$" by Ye et al. studies the inversions of urban $CO_2$ emissions based on $CO_2$ data from the OCO-2 mission. It analyzes both Observing System Simulation

10    Experiments and experiments with the real $CO_2$ data from the OCO-2 mission to assess the level of precision and accuracy that could be expected from such inversions.

This is a critical topic in a context of the preparation of various missions for the monitoring of $CO_2$ anthropogenic emissions from space while there is still a lack of studies addressing the feasibility of this monitoring in details. This study gathers tools and material that can feed the

15    understanding and discussions in this research field. I think that the authors can produce an important paper with such a basis.

However, I also think that the text needs a strong improvement and that critical issues need to be addressed:

1) Most of the text needs a careful rewriting. At first sight, it looks concise. But when reading it

20    into details, especially when trying to understand how some of the curious results (see below) have been obtained, it appears to be confusing, hasty, approximative, full of gaps, and unstructured. A major issue is the lack of logical flow and of robust focuses. We often do not know where the authors go, and why they make their choices. Large parts of the manuscript look like piles of independent pieces of explanations, analysis and discussions that have not been

25    linked or synthesized. Some of the comments below illustrate this. The second half of the abstract, sections 1 (the introduction), 4 and 5 give an illustration of the lack of logical structure. The second half of the abstract, sections 2.3 and 3.2 (among others) provide a good illustration of the lack of clarity, especially when dealing with the experimental protocol.

**Response**: We appreciate the referee very much for these comments. We have tried our best to

30    revise the manuscript carefully to make it clearer and better organized, especially focusing on the logical flow and linkage among all the sections and experiments in this paper. We re-organized the different test cases according to the major components of the problem, i.e. transport errors,

biogenic flux contributions, and background conditions. Our corresponding revisions are explained and summarized as follows.

- The experimental protocol and details of our methodology have been reformulated. Please see section 2 for details.

- The section of results (section 3) has been reorganized. The emission optimization using real OCO-2 data has been shown at the beginning of this section, in order to demonstrate the specification of background concentration and derivation of $X_{CO2}$ enhancements from observations. Based on the emission optimization using single tracks, we want to highlight the performance of the propagation method of the transport model errors and their impact on emission optimization.

- Other sections are all revised carefully focusing on the logical flow.

- Most of the texts have been re-written. Please see the responses to the following comments for further details.

2) I do not understand central results and figures of this manuscript. It may be because I do not really understand the details of the computations.

- section 3.2.a and Figure 8: I do not understand what is done when several tracks from different days are used together. Do the authors gather the corresponding 100 perturbed wind conditions together, forming a much bigger ensemble of perturbed wind conditions, and then look at the distribution of S arising from this larger ensemble? In that case, the convergence they show in Figure 8 would only be an illustration of the decrease of the sampling errors for the distribution of the transport error when increasing the sampling size, i.e., nothing about the asset of using different OCO-2 tracks. Or do the authors do things in such a way that N tracks bring N independent statistical transport errors together, which would decrease the overall weight of the transport error? Then, in the absence of prior and measurement error, the curve on Figure 8 should converge towards 0. My lack of understanding is a real concern since this is one of the main results that are cited in the discussion, conclusion and abstract.

**Response**: Thanks for this comment. We have made many revisions to explain our method in more detail, and try to delineate the uncertainty in scaling factor when using multiple tracks in a better way.

In this section, a Monte Carlo method is used to evaluate the impact of transport model errors on emission optimization. In this work, a scaling factor ($S$) is used to optimize the urban $CO_2$ emissions as a whole to get the best match between simulated and observed total $ffXCO_2$ enhancements for each track. The spatial structure of emissions is assumed to be correct and

remained the same. For the prior emission errors, we assumed no temporal correlation primarily because OCO-2 tracks occur at low frequency, with several weeks between tracks. Thus the scaling factor is calculated for each track separately. We did not include spatial information on prior emissions errors as current emission products (e.g. ODIAC) for most cities do not have an uncertainty estimate (section 3.2, P16, L9-12). For the measurement errors, we have worked with the Jet Propulsion Laboratory but the results are not available at this point for high-resolution spatial and temporal errors, as well as their correlation (section 4.3, P21, L35-36 and P22, L1-12). This work is ongoing and will be published in another manuscript when version 9 of the data is available. Since in this work the main focus is the impact of transport model errors, biogenic fluxes, and boundary conditions (background), in the OSSEs we illustrated these components assuming independent tracks, correcting for biases in fossil fuel emissions. These descriptions have been added.

For the OSSEs, assuming that there are $N$ tracks available (here $N$=1, …, 20) with pronounced fossil-fuel $X_{CO2}$ enhancements, the pseudo data and scaling factors are calculated as follows (see section 2.5.3):

i). Randomly select $N$ tracks of $XCO_2$ along a typical ground track of OCO-2 from all the pseudo observations over a city. The random selection gives an overall measure for different atmospheric transport conditions.

ii). Scaling factors ($\{S_n\}$, n=1,2,…, N) are calculated for each pair of pseudo observation and pseudo modeling data ($ffX_{CO2ff,o}$, and $ffX_{CO2ff,m}$). For "plume city", the random transport errors in wind speed and wind direction are incorporated by rescaling the plume for each pair of tracks. For "basin city", the scaling factor is assumed to follow a normal distribution determined by the results of ensemble modeling. The distribution average value and standard deviation are set up using the median and semi-full-range (half of the difference between maximum and minimum) of the scaling factors calculated using the 18 ensemble members. The methods are described in section 2.5.1.

iii). Since the distribution of scaling factor is often skewed due to the non-linearity of the transport model, the median value $\tilde{S} = median(\{S_n\}, n = 1,2, ..., N)$ for N tracks is used. The spread of $S_n$ is represented by the semi-interquartile range, which is defined as $R = (Q_3 - Q_1)/2$, i.e. half of the difference between the 75th percentile ($Q_3$) and the 25th percentile ($Q_1$) of $\{S_n\}$.

iv). The Monte Carlo method is used to derive the statistical distributions of $\tilde{S}$ and $R$. We repeat i), ii) and iii) for $K$ times ($K$=10³ in the revised manuscript, in original version $K$=100) to get all the scaling factors $\{S_n\}_k$, $\tilde{S}_k$ and $R_k$ (n=1,2,…,N, k=1,2,…,K).

In our previous manuscript, the median of $\tilde{S}$ and its standard deviation $\sigma(\tilde{S})$ was displayed in Figure 8, which showed decreasing $\sigma(\tilde{S})$ with increasing number of tracks. The repeated random selection of tracks ensures that different conditions of atmospheric transport are included, therefore there is no sampling error in $\sigma(\tilde{S})$, which only represents the impact of transport model errors. Since the transport model errors are assumed to be unbiased, and the prior emissions errors and measurement errors are not included, the scaling factor is expected to converge to its truth (i.e. 1). However, we admit that the uncertainty spread related to transport model errors could be underestimated by only $\sigma(\tilde{S})$, which was probably the primary reason for the too optimistic results in our manuscript previously. It was also pointed out by our Referee #1. Therefore, in the revised manuscript, the semi-interquartile range (half of the difference between $25^{th}$ to $75^{th}$ percentiles) of $\{S_n\}$ (n=1, 2,…,$N$) is used to measure the spread of the scaling factors for $N$ tracks.

Figures 10, 11, and 12 in the revised manuscript show the probability density of $\tilde{S}$ and $R$ for different number of tracks ($N$). The results also show convergence in the median scaling factor and the semi-interquartile range. The number of tracks to maximize the performance to constrain the scaling factor uncertainty related to transport errors is estimated at a minimum of 10 for Riyadh, Cairo and LA. Please refer to section 3.2 for the detailed analysis.

- section 3.2.b: an OSSE is conducted with synthetic perturbations of the transport. I cannot understand how the bias in the results can be related to a bias of WRF in the real world. My very simple understanding is that this bias is just a consequence of the way the perturbations are chosen and applied in the synthetic world, combined with the non-linearity of WRF. Again, this corresponds to one of the main results in the abstract, discussion and conclusion.

**Response**: Thanks for this comment. Based on the statistical results from previous studies (e.g. Feng et al., 2016), the simulated surface wind speeds by WRF model is usually overestimated over the Los Angeles basin (for example), where low wind speed or stagnant transport conditions happen more often than in "plume" cities. Therefore, in the experiment for Los Angeles, we assumed that the wind speed errors ($\varepsilon$) are positively biased. In the real world, the larger simulated wind speeds result in stronger diffusions of fossil-fuel $CO_2$ in downwind direction of the emission sources, which can lead to smaller $X_{CO2}$ enhancements in the modeling results, and therefore larger emission scaling factor (positively biased).

In addition, the OSSEs for plume cities (Riyadh and Cairo) show that, the scaling factor ($S$) converges to truth, i.e. one, with unbiased transport model errors, although the statistical distributions of $S$ calculated using local $X_{CO_2}$ enhancements derived from real OCO-2 data and the simulated ff$X_{CO_2}$ with transport model errors are skewed or even bimodal (see Fig. 7). The statistics of transport errors are based on a published study for Indianapolis, IN, which is an inland city with similar terrain (Deng et al., 2017). In past study, it has been shown that flat terrain does not produce any biases in wind speed nor direction, contrary to basin city where topography plays a major role.

Transport by the atmosphere is a linear process but the intersect between the plume and OCO-2 track can create non-linear problems, for example the track and the plume makes a variable angle between them, due to atmospheric transport. Small errors in wind direction increase rapidly when propagated into the emission scaling factor when the angle is small (when the track detects the very edge of a plume). We applied a threshold of the angle ($10°\sim170°$) to avoid these situations in the pseudo-data experiment (transport error calculation) but the real-data examples used here are still affected by this problem.

With the ensemble approach, the overall wind speed bias of the ensemble average results is reduced to 0.48 ms$^{-1}$ when compared to observations over the entire domain. The impact of this wind bias is taken into consideration by using a factor (k) to further dilute or increase the simulated ff$X_{CO_2}$ along the track set up in the OSSE for LA at each simulation output time, where:

$$k = \frac{\bar{u}}{\bar{u}+\Delta},$$

$\bar{u}$ is the average wind speed over the domain, and $\Delta$ is the corresponding bias in m s$^{-1}$.

These above descriptions are added in the revision (section 2.5.1) to clarify the ensemble modeling and the method to account for the positive bias in wind speed in the OSSE for LA.

**References**

Feng, S., Lauvaux, T., Newman, S., Rao, P., Ahmadov, R., Deng, A., Díaz-Isaac, L. I., Duren, R. M., Fischer, M. L., Gerbig, C., Gurney, K. R., Huang, J., Jeong, S., Li, Z., Miller, C. E., O'Keeffe, D., Patarasuk, R., Sander, S. P., Song, Y., Wong, K. W. and Yung, Y. L.: Los Angeles megacity: a high-resolution land–atmosphere modelling system for urban CO2 emissions, Atmos. Chem. Phys., 16(14), 9019–9045, doi:10.5194/acp-16-9019-2016, 2016.Deng, A., Lauvaux, T., Davis, K. J., Gaudet, B. J., Miles, N., Richardson, S. J., Wu, K., Sarmiento, D. P., Hardesty, R. M., Bonin, T. A., Brewer, W. A. and Gurney, K. R.: Toward reduced transport errors in a high resolution urban CO2 inversion system, Elem Sci Anth, 5, 20, doi:10.1525/elementa.133, 2017.

- section 3.4 and Figure 11: the text at lines 11-12 page 11 says something really strange for me: "bimodal distributions for three tracks due to negative perturbed wind speeds caused by large errors added on small absolute wind speed". What is a negative perturbed wind speed? What do the authors do with such a wind speed? Furthermore, they are a lot of negative S in Figure 11 (the

5    text does not speak about it). How can it be possible? Would that mean that the authors diagnose negative enhancements of $XCO_2$ along the simulated or observed OCO-2 tracks? I would not understand it but I do not really know how the authors diagnose the enhancements since section 2.3.1 and the corresponding equation on page 6 lack of rigor. Is there a link between these negative scaling factors (turning cities into sinks of $CO_2$) and the "negative perturbed wind speed"?

10   **Response**: Thanks for this comment. Sorry for the mistake in the texts in this section. The bimodal distributions of the scaling factor for some of the tracks are related to the non-linearity in the transport errors. The negative values of scaling factor can be explained by our method of transport model errors propagation for the "plume cities", which has been detailed in the revised manuscript (section 2.5.1) and the supplement. In brief, with the assumptions of 1) constant wind

15   speed coinciding with x direction, 2) negligible turbulent diffusion in x direction compared to advection, and 3) constant eddy diffusivity in y and z directions, the atmospheric diffusion equation would be a linear partial differential equation (S1), which can be non-dimensionalized as equation S2. The equation shows that, the non-dimensional $CO_2$ concentration is independent of wind speed, eddy diffusivities, and surface fluxes. With these assumptions, and given the

20   numerical solution of an $XCO_2$ plume by simulations, the $X_{CO2}$ distribution under perturbed wind speed $\bar{u} + \varepsilon$ can be estimated by transforming the simulation results as: $\overline{c_{r\prime}}(x_{r\prime}, y_{r\prime}, t) = \overline{c_r}(x_r, y_r, t)$, where $x_{r\prime} = \frac{\bar{u}+\varepsilon}{\bar{u}}(x_r - x_0) + x_0, y_{r\prime} = y_r, z_{r\prime} = z_r, \overline{c_{r\prime}} = \bar{c_r}\frac{\bar{u}}{\bar{u}+\varepsilon}$. Here $\bar{c_r}(x_r, y_r, t)$ is the simulated $X_{CO2}$ distribution in model domain when incorporating wind direction error of $\theta$ in degree, which is derived by rotating simulated plume by the angle of $\theta$ about the emission

25   center $(x_0, y_0)$.

     This method is a tradeoff between Gaussian plume and realistic $X_{CO2}$ distribution, which would need huge computational cost to use an ensemble method instead of error propagation through plume perturbations. The definition of $x_0, y_0$ is described in the supplement, please refer to it for more details. Here the wind speed error is assumed to follow a normal distribution of N(0,

30   1) (unit: m s$^{-1}$). When the wind speed error is too negative, one would have $\bar{u} + \varepsilon < 0$, thus the transformed $X_{CO2}$ ($\overline{c_{r\prime}}$) would be negative. This is unrealistic since the anthropogenic $X_{CO2}$ should always be positive. Because we wanted to keep all the cases in the histogram of scaling factor corresponding to the wind speed error distribution, the negative values are kept in the previous

manuscript. To ensure appropriate values of the transformed $X_{CO2}$, in the revised manuscript, all the cases of random wind speed errors that lead to $\bar{u} + \varepsilon < 0$ have been omitted. Please see the revised histogram of the scaling factor in Fig. 8, and the corresponding analysis in section 3.1.

5    3) The study of the uncertainty in the ecosystem fluxes in the PRD area does not fit with the rest of the paper and the authors did not make much effort to connect the results and discussions corresponding to this study to the other ones. I think that this should be removed, which would help improve the structure and logical flow of the paper. We get a strong feeling that the authors wanted to value an experiment they had done in the PRD region by inserting it artificially into an

10    independent paper. It is all the more problematic that the dissymmetry between the experiments in the "plume cities" and LA is already a bit difficult to admit (with these two cases only, we can already get the feeling of following two independent studies in parallel). There is no justification for not applying the range of tests on the transport errors to the PRD area. And there is no reason for not applying the tests applied in the PRD area on the biogenic fluxes to, at least, LA, and,

15    maybe, Cairo. Using VPRM in LA when assimilating real data vs. using an ensemble of larger scale products for the OSSEs in the PRD area (i.e., different types of tools) emphasize the feeling that the PRD sections are disconnected from the rest. At last, I think that the analysis in 3.3 are unclear and debatable so it would have to be extended and improved. On a similar topic: the oscillation of the text between distinguishing or mixing the background and biogenic components

20    is problematic, especially since the background is defined several times in different ways in this paper. This, in addition to the problem stated above, leads to strange sentences at the end of the paper ("biogenic fluxes are critical for cities located in well vegetated areas . . . Background mole fractions of $XCO_2$ for urban areas require more consideration . . . More sophisticated biospheric modeling can help to develop a better determination of the background $XCO_2$").

25    **Response**: Thanks for this comment. We have revised the structure of this manuscript to clarify our outline and the connection between sections (see section 1, section 3.1, 3.2, 3.3, and section 4). This work is focused on the quantification of fossil-fuel $CO_2$ emissions using spaceborne measurements of $XCO_2$. In this manuscript, the major research question is to assess the feasibility of doing this from two aspects of this emission attribution problem. Firstly, the question is

30    simplified by ignoring the biospheric contribution to the local $XCO_2$ enhancements, or assuming that the biospheric signals can be simulated very well. Synthetic data are used in this section to evaluate impact of transport model errors on the emissions estimates, which is scaled as a whole for a city. The "plume cities" (Riyadh and Cairo) and "basin city" (LA) are shown with different

transport model error propagation methods, due to their different features of fossil-fuel $XCO_2$ enhancements with/without the confinement of local topography. For "plume cities", we used a rescaling method to propagate the transport model errors. For "basin city", the rescaling method would not work due to terrain effect, thus we used ensemble modeling to estimate the uncertainty

5    spread of fossil-fuel $XCO_2$ enhancement.

The assumption of well disentangled fossil-fuel $XCO_2$ signals is only valid for isolated cities in a region that barely impacted by local and regional biogenic carbon fluxes. In most cases, the anthropogenic $XCO_2$ enhancements along a satellite track would be obscured by biogenic signals due to the spatial inhomogeneity of biogenic $CO_2$ fluxes and regional advection. The

10    disentangling of fossil-fuel signal from biogenic signals has been discussed in other studies using flask observations, CO measurements, etc (e.g. Turnbull et al., 2015). However, the uncertainties in the biogenic signals and the associated uncertainty in emission optimization have not been fully addressed for satellite data along tracks in previous studies. Therefore, we conducted the simulations for the Pearl River Delta (PRD) region, aiming at revealing the impact of local

15    biogenic fluxes on extracting local fossil-fuel $XCO_2$ enhancements.

Admittedly, the manuscript would be better organized with all experiments conducted for the same cities. However, we would like to show cities with different complexity, which is critical for addressing the emission monitoring for most cities around the globe. The PRD area is chosen because the cities are located in a densely-vegetated area, which is a typical example for biogenic

20    contribution. Since the biogenic contributions are very important for many cities around the globe, we would like to keep this section, with many revisions made. The PRD example is shown here to indicate the potential impact of biogenic fluxes, and how the urban-to-rural vegetation distribution gradient can affect our ability to estimate anthropogenic emissions using satellite data. A second manuscript in preparation will include a broader description of vegetation signals for

25    many cities in a more general context.

The revisions are summarized as follows:

i).    To justify the selection of cities for experiments with different vegetation coverage, the land use information is added in Table 1, based on MODIS IGBP 21-category data. The land cover categories accounting for more than 90% in total of the innermost domain are

30        listed. For the selected cities, we also derived the local biospheric fluxes (NEE) from MsTMIP data. In order to show the local variability of NEE, we calculated the difference of NEE averaged between the urban land cover and non-urban grids. The figure for the PRD region has been added in the supplement (Fig. S2) for reference.

ii). The biospheric contribution to local $X_{CO_2}$ enhancement is an important concern for cities located with local or remotely impact of biospheric activities. It is crucial for many cities around the globe, which could obscure the interpretation of observations. The introduction and discussion sections (section 1 and 4) have been revised to highlight this problem (P4, L20-29; P20, L23-32).

iii). The specification of background $X_{CO_2}$ has been clarified for extracting the $X_{CO_2}$ enhancements related to local fluxes from OCO-2 retrievals. A section (section 2.2) has been added. For consistency, we used the same method to specify background for each track in the revised version. The background $X_{CO_2}$ determined using VPRM simulation for LA is removed. Generally, for in-situ measurements, we need to measure $CO_2$ at an upwind site (background) and a downwind site, assuming a same air parcel is measured. Therefore, the $CO_2$ concentration difference at the two sites can be used to represent influence form local fluxes. Typically it is required that the two stations are located along the wind direction, which is usually not the case for satellite observations. Thus, we can expect that a constant background is inappropriate for satellite measurements along tracks. Here a background line is used for each track, which is determined for each track by using a linear regression of $X_{CO_2}$ versus latitude. For each track, the OCO-2 soundings used for the regression are selected from all available 1-s averaged $X_{CO_2}$ within the latitudinal range of interest for each city, with a criterion of $X_{CO_2} < \mu + 0.5\sigma$, where $\mu$ and $\sigma$ are the mean and standard deviation of the detrended data. The results in the revision show better comparison of the simulated and observed local enhancements, which indicates the advantage of the linear background. Please see section 3.1 for more details.

iv). The analysis in section 3.3 has been extended and improved to better illustrate the biogenic influences on the local $X_{CO_2}$ enhancements. The biogenic contribution for LA has also been discussed in section 3.3. We note that there are two reasons for the difficulty of implementing a simulation for LA similar to that for the PRD region. First, in the simulations for the PRD region, the NEE maps are downscaled using the green vegetation fraction (GVF). However, the default MODIS-based GVF provided with the WRF model over LA shows nonrealistic value and structures when compared with the real-time MODIS-based GVF maps derived by Vahmani and Ban-Weiss (2016), which is shown in the supplement (Fig. S7). The update of GVF data is out of scope of this study. Second, the NEE maps are downscaled assuming the constant vegetation productivity within a signle NEE grid cell (0.5 °×0.5 °). Los Angeles has a variety of climate zones because of its proximity to the Pacifc Ocean and the nearby mountain ranges, where a

variety of vegetation species exist with different growth patterns (McPherson et al., 2008). Thus, it could be nonrealistic to assume the constant productivity in a grid cell and disregarding the classification of climate zone. More comprehensive data and method are needed to fulfill the estimation of biospheric contribution to local $X_{CO2}$ enhancements by simulations.

**References**

Vahmani, P. and Ban-Weiss, G. A.: Impact of remotely sensed albedo and vegetation fraction on simulation of urban climate in WRF-urban canopy model: A case study of the urban heat island in Los Angeles, J. Geophys. Res. Atmos., 121(4), 1511–1531, doi:10.1002/2015JD023718, 2016.

10    McPherson, E. G., Simpson, J. R., Xiao, Q. and Wu, C.: Los Angeles 1-Million Tree Canopy Cover Assessment. General Technical Report PSW-GTR-207, Albany, CA, 2008.

Turnbull, J. C., Sweeney, C., Karion, A., Newberger, T., Lehman, S. J., Tans, P. P., Davis, K. J., Lauvaux, T., Miles, N. L., Richardson, S. J., Cambaliza, M. O., Shepson, P. B., Gurney, K., Patarasuk, R. and Razlivanov, I.: Toward quantification and source sector identification of fossil fuel CO2 emissions from an urban area: Results from the INFLUX experiment, J. Geophys. Res. Atmos., 120(1), 292–312, doi:10.1002/2014JD022555, 2015.

4) The assumption that a change in the wind speed can be simulated by stretching the $X_{CO2}$ images and rescaling the concentrations needs to be discussed and justified. I think that this

20    assumption is not obvious and that is does not perfectly fit with many model formulations, such as a large range of Gaussian ones.

**Response**: Thanks for this comment. In brief, this error propagation method is a tradeoff between Gaussian plumes and simulations of realistic $X_{CO2}$ distributions with perturbed winds using WRF model, which would cause a huge computational cost. Actual studies use Gaussian plume models

25    to propagate errors, which we believe is an over-simplification of the transport problem. Instead, we applied known model errors from previous studies (e.g. Deng et al., 2017) to 1-km WRF simulations. We based our error propagation on the Gaussian plume model, but retained the plume structures by high-resolution simulations.

     With the assumptions for the Gaussian plume equation, the diffusion equation would

30    become a linear partial differential equation (PDE). The non-dimensional solution of that equation with different wind speed are the same, when assuming that the eddy diffusivities are constant. In other words, the Gaussian plume under perturbed wind speed can be retrieved by

rescaling the original plume following the non-dimensional scaling (please refer to the supplement for details). In this work, this feature is applied to the simulated $X_{CO2}$ distributions, i.e. representing the random errors in wind speed and wind direction by rescaling the simulated $X_{CO2}$ distribution. This method helps to retain the complexity of realistic distributions of ff$X_{CO2}$

5    compared to using Gaussian plumes. The reason is that, Gaussian plumes are more suitable for estimation of long-term average distributions of concentration, but the satellite observations are snapshots of $X_{CO2}$, since it takes only a few tens of seconds for OCO-2 to fly over a city. Thus the high-resolution forward modeling is a better approach to reproduce the urban ff$X_{CO2}$. However, the forward simulations would lead to a much larger computational cost, compared to using a

10    Monte Carlo method to represent random transport model errors and selection of tracks. Therefore, we compromised to use the rescaling method for this purpose.

These above descriptions have been included in the supplement.

*References*

Deng, A., Lauvaux, T., Davis, K. J., Gaudet, B. J., Miles, N., Richardson, S. J., Wu, K., Sarmiento, D. P.,

15    Hardesty, R. M., Bonin, T. A., Brewer, W. A. and Gurney, K. R.: Toward reduced transport errors in a high resolution urban CO2 inversion system, Elem Sci Anth, 5, 20, doi:10.1525/elementa.133, 2017.

5) The paper focuses on transport, biogenic and background errors. This justifies the lack of discussion on the measurement errors in the sections dedicated to the OSSEs. However, these

20    measurement errors are nearly ignored when analyzing the results with real data, in the discussion, in the conclusion and even in the introduction (e.g. on p3), while other studies indicate that this is a critical component of the problem. As a consequence, the presentation of the conclusions from section 3 often seems biased and misleading. E.g., lines 11-18 p14 (but also lines 18-19 p1 in the abstract) do not really say that the 15%/5% numbers relate to transport errors only and that the

25    potential of the OCO-2 data themselves (i.e. not that of the modeling framework) is not fully investigated. This could strongly favors over-optimistic reading and citations of this paper (such as when this paper itself summarizes in a very optimistic way the conclusions from Hakkarainen et al. 2016 on p3 line 3). In such a context, I also think that the title of the manuscript itself is misleading.

30    **Response**: Thanks for this comment. The 15%/5% numbers are indeed related to only the uncertainty in transport that we can expect when using abundant number of tracks. The focus of the OSSEs is to show a computational efficient method to propagate transport model errors to the

ffX$_{CO2}$ in simulation results. In terms of the measurement errors at urban scale, discussion about the measurement errors has been added in section 4 (P21, L35-36; P22, L1-12), based on some recent studies on the uncertainty of the OCO-2 X$_{CO2}$ data. The largest uncertainty in OCO-2 X$_{CO2}$ data has been suggested to be related to the interferences of surface properties (such as surface pressure or albedo) and aerosols. However, currently the nonlinearities in the retrieval or random components of interference error are likely poorly characterized. It's still difficult to suffice to characterize potential errors at high resolution and their fine scale structures. We have been discussing this issue with the OCO-2 Science Team and will provide a more detailed analysis of fine-scale errors in the future.

The limitations corresponding to the measurement errors of the OSSEs' results have been highlighted in the abstract, section 4 (discussion) and section 5 (conclusions). Please refer to these sections for details. In brief, we note that the above OSSEs cannot demonstrate the actual performance of OCO-2 data directly, but aims at evaluating the potential of high-resolution inversion techniques with the imperfection of the atmospheric transport modeling, the biogenic fluxes, and the background conditions. For a more comprehensive use of OCO-2 measurements, a sophisticated inversion framework incorporating the errors in measurements and prior emissions along with the error correlation structures should be developed in future work.

For the title of the manuscript, we decide to keep the current title. A similar title is also used for a study on the potential of CarbonSAT data for inverse estimates of urban emissions based on OSSEs (Pillai et al., 2016).

Pillai, D., Buchwitz, M., Gerbig, C., Koch, T., Reuter, M., Bovensmann, H., Marshall, J. and Burrows, J. P.: Tracking city CO2 emissions from space using a high-resolution inverse modelling approach: a case study for Berlin, Germany, Atmos. Chem. Phys., 16(15), 9591–9610, doi:10.5194/acp-16-9591-2016, 2016.

6) Section 4 does not deliver the type of discussions that are expected when reaching the end of section 3. Section 3 does not go really deep into the result analysis, in particular when looking at the tests with real data (while these tests will certainly be brought forward when citing this paper). Therefore, section 4 should provide more insights on the results. However, the first paragraph in section 4 is merely (if we except 2, 3 lines) a summary of what has already been said in section 3, and the rest of section 4 is the piling of small and independent discussions on the perspectives.

**Response**: Thanks for this comment. The analysis and discussion sections have been all revised and re-written. In particular, the real data analysis has been moved to section 3.1, with more

explanations about the objective of the real data test, as well as the insights about thw interpretation of local $X_{CO_2}$ measurements that can be drawn from these results. The discussion section (section 4) has been revised to suggest the limitations of the results related to our selection of cities (section 4.1), major challenges about the interpretation of observations for most of the

5   global cities (section 4.2), and the insights drawn from the OSSEs' results and the limitations (section 4.3). Please refer to these sections for details.

7) I could list a lot of minor issues. However, nearly all of them relate to comment 1 and I will thus wait for a later step before undertaking a detailed list. That said, I would like to mention:

10  - that the references could be complemented by Schwander et al. 2017 (Schwandner, F. M. et al., Spaceborne detection of localized carbon dioxide sources, Science, 358 eaarn5782, 2017) which could potentially impact lines 4-5 p3. Nassar et al. 2017 (Nassar, R., et al., Quantifying CO2 Emissions From Individual Power Plants From Space, Geophysical Research Letters, 44, 10045-10053, 10.1002/2017GL074702, 2017) is also relevant for the discussion on page 3 since power

15  plants and cities are sometimes considered as similar targets and since the quantification of their emissions using OCO2 data raise similar challenges.

**Response**: Thanks for this suggestion. These references have been added in our manuscript in the introduction (section 1, P3, L32-33).

- section 2.2.1 and table 1: the information about the spatial resolution of the outer domains is not

20  really interesting if there is no information about their spatial extent.

**Response**: The outer domain is critical for atmospheric boundary conditions, almost systematically included to document the WRF configuration and demonstrate the validity of the model simulations. For this particular study, we have removed the information about the outer domain from the main text and Table 1, and just described the innermost domain. Instead, the

25  domain location and emission distribution for the selected cities are added to the supplement (Fig. S3).

- regarding the integration of the data along the OCO2 tracks: as said before, the section 2.3.1 and its equation are unclear; I did not understand what is the spatial and/or temporal representativity of the dots in figure 5; and I am not sure about the meaning of the spatial location of the OCO2

30  dots in figure 6.

**Response**: In this work the OCO-2 data are firstly filtered by Quality Flag (QF=0), then averaged using a 1-s time window to smooth some observation noise. The latitude and longitude coordinates are averaged accordingly to get the geolocation of the averaged sounding. Then we selected the 1-s average $X_{CO2}$ by geolocation, i.e. selecting soundings located within the domain of interest (innermost simulation domain). Note that a very large domain with spatial resolution of 4 km is used for the forward simulations for Los Angeles (see Table 1), therefore, a smaller domain is used as the domain of interest with the longitude limit of 119.0 °W~116.3 °W and latitude of 32.2 °N~35.7 °N. The data selected with the above methods are integrated by latitude along each track. In Figs.4-7, all the selected tracks are shown for each city, along with the simulated ff$X_{CO2}$. The comparison and emission optimizations are also revised (section 3.1).

- The paper often considers the temporal variability of the emissions as a side problem while one of the central tests studies the asset of using several satellite tracks. The day to day variability of the emissions is merely ignored. The seasonal variations are mentioned but they are not confronted to the statements regarding the number of OCO2 tracks required to balance the transport errors, nor to the actual number of available tracks per year.

**Response**: Typical day-to-day variations are small except for weekly emission patterns, such as week days and weekends. Diurnal variability can be much larger, but OCO-2 overpass remains nearly the same time of day. ODIAC includes monthly emission estimates, which describes the seasonal variations. When using multiple tracks of data to constrain the emissions, we have been assuming negligible day-to-day variations in the total emission over a city. Real cases include monthly variability. For pseudo-data, the scaling factors calculated for the multiple tracks with only the transport model errors propagated, the true scaling factor would be one. Since we focus on the impact of transport model errors in this work, neglecting the prior emission errors will have a minor impact on our conclusions about the contribution of multiple tracks on constraining the uncertainty associated with transport errors. These texts have been added (section 3.2, P16, L9-19).

Day-to-day variations in biogenic fluxes are more important for future inversions, as well as diurnal emission cycles for satellite missions sampling at different times. We agree that a more sophisticated inverse modeling framework for multiple tracks should be developed in future work.

- p7 : lines 29-31: i) due to the variations of the wind as a function of the vertical (including wind curl in the PBL), modifying the PBL should have an impact on the XCO2 fields ii) if the PBLH and vertical mixing near the surface do not have much impact, why having perturbed the PBL schemes in these experiments?

**Response**: Because we use a mesoscale model, PBL schemes control not only the PBL depth but also the wind conditions and most meteorological variables near the surface. The PBL depth has a minor impact on our results thanks to the dilution of the signals across the atmospheric column. However, the wind speed and direction are controlled primarily by the near-surface dynamics, simulated by the PBL scheme. Therefore we used different PBL scheme to carry out the ensemble modeling.

- p11 l 32 If I am not wrong, such a scaling was not applied for Cairo and Riyadh: why? how to rely in this factor 1.288 while the inventory are assumed to be uncertain? why not just acknowledging that the estimations correspond to the satellite track times and discussing the extrapolation into daily budgets later?

**Response**: Thanks for these questions and suggestion. In the revision, we included the discussion of the contribution of temporal variability on the emission optimization for Riyadh, Cairo, and LA in discussion (section 4.3). The effect of temporal sampling bias due to daytime overpass of the satellite has been highlighted, which can lead to an overestimation of the emissions estimations (P22, L36-37; P23, L1-7) if this sampling bias related to diurnal variations of the emissions is ignored.

- section 3: I feel that the results when using real data (especially in LA) raise severe doubts regarding the potential of the inversion strategy or of the current OCO-2 data, and that this is not highlighted in the conclusion and abstract of the paper. At least, this deserved more discussions in section 4.

**Response**: Thanks for this comment. With the revised specification of background concentration, the results show better comparison of the local observed $X_{CO2}$ enhancement and the simulated ff$X_{CO2}$ (section 3.1.1 and 3.1.2). The discrepancies between the observed local enhancements and simulated ff$X_{CO2}$ have also been discussed in section 3.1.1 (P13, L31-36; P14, L1-9), which is associated with the measurement errors, imperfection of the specification of background $X_{CO2}$, prior emission errors, in spite of transport errors. For LA, the measurement errors in the OCO-2 data may be also subject to errors due to pointing errors of different instrument bands. This can be seen from the high warn levels of the data over the northern and southern edges of the desert north to LA, which is shown in the Fig. S6 (in the supplement). The OCO-2 Science Team is currently working on an updated version of the data set which could improve the measured enhancements over steep terrain.

- there are a lot of awkward sentences throughout the text such as "These cycles could be compensated by optimal sampling strategies but only active sensors will be able to sample across clouds and at night. For future missions, the sampling bias might be compensated by more frequent tracks or targeted view modes." at the end of section 4.

5   **Response**: We apologize for the confusing expressions in this sentence. Actually we would like to highlight a shortcoming of OCO-2 measurements of $XCO_2$, which is merely available in daytime and can only provide constraints of emissions for limited temporal range. In perspective of seasonal data availability, the data yield is associated with variations of cloud cover, which blocks the sensor from observing. We expect that these temporal "blind" observation periods

10   could be covered by deploying active sensors measuring in both day and night. Meanwhile, we expect that the satellite observations targeting specific locations, e.g. OCO-3 and GOSAT, could provide more frequent sampling in areas with larger difficulty due to more frequent cloud contaminations. This will facilitate the monitoring of major anthropogenic emission sources around the globe. The explanations have been added (section 4.3, P22, L25-37; P23, L1-7).

**RC2 by Anonymous Referee #3**

This study investigates the potential of observations of total column CO2 ($XCO_2$) from a satellite of type OCO-2 to quantify $CO_2$ emissions from individual cities. The study is based on high -resolution (kilometer scale) urban plume simulations with WRF-CHEM used to conduct observation system simulation experiments and for comparison with real OCO-2 observations.

The topic is highly relevant and timely as several new $CO_2$ satellite missions are currently being planned and more and more applications of OCO-2 data are being publishes. The manuscript is thus a welcome contribution.

The manuscript is fairly well written and the methods, centered around comprehensive model simulations, are generally sound. The study presents many interesting and innovative aspects that clearly deserve being published. In particular, the study investigates uncertainties in the emission estimates related to transport uncertainties by stretching/squeezing or rotating the simulated model fields, applies an ensemble approach for basin cities, and investigates the influence of biospheric fluxes using an ensemble of biosphere flux models downscaled to high resolution for the simulations. The simulation setup is impressive and the analyses presented are comprehensive.

Despite these positive aspects, I also have a few concerns primarily related to the way the study is presented and the conclusions that are drawn as detailed in the following.

Main points:

Based on the abstract, a reader may conclude that OCO-2 is a satellite highly suitable for quantifying $CO_2$ emissions from cities, but OCO-2 has not been designed for this purpose and is clearly far from ideal. The main problem of the manuscript is that the severe limitations of OCO-2 in terms of temporal and spatial coverage are not clearly discussed and that, therefore, a much too optimistic picture is drawn of what can be achieved with such a satellite. The only sentence in the introduction addressing the issue of coverage is the following fairly neutral statement: "discernible $CO_2$ emission imprints can be limited due to the contamination by clouds and aerosols and limitations of spatial-temporal sampling coverage for local sources related to the revisit cycle of sun-synchronous polar orbit and the narrow tracks". OCO-2 has about 15 orbits per day, each with a swath of approx. 10 km. In 1 day it thus covers a total east-west extent of 150 km. For comparison, the circumference of the Earth is about 40'000 km, i.e. OCO- 2 would take 266 days to sample each point on the globe (at the equator) at least once. But OCO-2 has a 16 days repeat cycle which means that many points on the globe will never be observed at all. To

measure the plume of a city does not necessarily require flying directly over the city (except for basin cities), but the overpass should be close to have sufficient signal and to be able to unambiguously attribute the plume to its source. On page 2, L35, the authors state that "OCO-2 pioneered the contiguous high-resolution mapping of global $CO_2$ concentrations", which needs to be changed. A swath of 8 pixels across-track definitely does not qualify OCO-2 as a "mapping mission". OCO-2 takes high-resolution measurements along a narrow line. Its sampling strategy is much closer to a 1D than a 2D mission, which is also why this study analyzes $CO_2$ along individual (1D) lines and not in (2D) images of entire plumes. It is important to make this distinction because several true imaging missions are currently being planned including geostationary and polar orbiting missions.

There are also other reasons why the overall tone of the manuscript is much too optimistic: On page 3 the manuscript states that "a handful of cities with different typical $XCO_2$ features in $XCO_2$ff enhancements" were selected, which gives the impression that the selection was more or less arbitrary and that any other combinations of cities might have worked equally well. However, the authors have picked highly ideal cities with a) very large emissions, b) little interference with biospheric fluxes (Riyadh and Cairo), c) very low average cloud cover, and d) nicely isolated from other cities avoiding overlapping plumes. For the study of interferences with the biosphere, the Pearl River Delta region has been selected, one of the most densely populated regions where, again, anthropogenic emissions are unusually large compared to biospheric influences. There are good reasons for selecting these cities because OCO-2 offers many opportunities to observe their plumes, but it has to be openly communicated why these were selected and that the challenges for most other cities (probably for 99% of all cities of the globe) will be much larger due to frequent cloud coverage, strong interferences with biospheric fluxes (especially during summer coinciding with periods of low cloud cover and hence good observation opportunities), poor coverage due to the OCO-2 orbit geometry and narrow swath, overlapping plumes, plumes below the detection limit etc. Based on the OSSEs it is concluded that emission uncertainties are constrained to less than 15% with at least 9-10 tracks for plume cities and even down to 5% for a basin city, and that it would only take about 2.1-2.4 years to collect a sufficient number of tracks with an OCO-2 type instrument. However, this conclusion is only valid for the unrealistic case of negligible observation uncertainties, perfectly known background and emission distributions, and non-existence of clouds. In reality, the magnitude of the plumes will typically be in the low ppm to sub-ppm range and hence in a similar range as instrument noise, and clouds will frequently obscure the view. Fitting real observations to the simulations is thus much more difficult and many more overpasses will be needed to reach such a low uncertainty. This is also nicely

demonstrated by the real cases presented in Section 3.4, where three overpasses over Riyadh provide median emission scaling factors differing by almost a factor of two (between 1.58 and 2.94), and the histograms even include negative emissions. These real OCO-2 observation cases would offer a nice opportunity to place the previous theoretical OSSE analyses in context and to
5    explain the additional challenges, but there is no discussion of this at all.

Thus, my general recommendations are

a) better emphasize the challenges as well as the limitations of OCO-2

b) stress clearly in the abstract and conclusions that the OSSEs were conducted under idealized conditions neglecting instrument uncertainties and that the convergence to XX% with YY tracks
10   only refers to the contribution of transport uncertainties under these conditions

c) explain that the cities selected in this study were chosen for their ideal properties to demonstrate the potential of OCO-2, but at the same time point out that for many other cities it will remain a challenge to quantify urban emissions with sufficient accuracy.

d) place the results obtained for the real cases in context with the OSSEs.

15   **Response**: Thanks for the above comments, which are very helpful and important to the objective expression of the results in this manuscript. The manuscript has been revised to address these problems, which are summarized as follows:

     a)   The challenges and limitations of OCO-2 on quantifying ffCO$_2$ emissions from cities are highlighted in the introduction (P4, L4-29), and discussion section 4.1 and 4.2, including
20          the obstacles due to spatial sampling coverage, detectability of ffX$_{CO2}$, and contaminations by clouds/aerosols (P19-20).

     b)   The limitations of the results of the OSSEs have been further clarified in section 3.2 and section 4.3. The limitations are related to the idealized condition of the OSSEs, i.e. considering cities isolated from other large cities nearby, and with weak interference with
25          biospheric fluxes; only transport model errors are incorporated in the OSSEs.

     c)   The selection of cities have been described in section 2.3 and discussed in section 4.1, including the fact that the cities (Riyadh, Cairo and LA) are chosen considering its ideal location, i.e. isolated from other large cities nearby, and weak contribution of biospheric signals. The challenges for many other cities are discussed in section 4.2, and emphasized
30          in the conclusions (section 5, P24, L31-37; P25, L1-7). With the rapid urbanization, some cities are located in conjunction with other sources nearby. Challenges still exist for most of the cities around the globe, associated with the limited detectability of imprints of

ffCO$_2$ emissions in XCO$_2$ due to cloud coverage, the OCO-2 orbit geometry (narrow swath and long revisit cycle), and plumes from cities with relatively weak XCO$_2$ enhancements below the detection limit.

d) The tests of real OCO-2 cases have been shown before the OSSEs in section 3.1. The results of these tests are revised, in order to show the local X$_{CO2}$ enhancements and detectability of these signals, as well as the impact of transport model errors on emission optimization when using a single OCO-2 track. These results are included in the context of the results from the OSSEs, as well as the section of discussion to emphasize the challenge. Meanwhile, following the comments by the other anonymous referee, we have revised the manuscript to better describe our modeling and error propagation method (section 2.5.1), along with the updated results for the specification of background concentration (section 2.2), which yield to better agreement between the observed and simulated X$_{CO2}$ enhancement (section 3.1).

Finally, I would like to point out that Figure 10 can not be published in its current form, since the lower part of the figure has been borrowed from another publication most likely violating copyrights.

**Response**: Thanks for this comment. We decide to remove this figure in the revised manuscript, because the condition of distribution of vegetation, and the corresponding contribution is more complex than that shown in this figure, which is quite associated with local transport conditions when we examine the modeling results for the Pearl River Delta region under different conditions of local wind field.

Small issues:

Page 1, Line 21: Change "in urban and rural area of Pearl River Delta" -> "in the urban and rural area of the Pearl River Delta"

**Response**: Done.

P2, L2: Although 70% of global CO2 emissions may be due to energy consumption in cities, a considerable (yet probably unknown) fraction of these emissions do not occur inside the cities but outside in power plants delivering the energy for the cities. This is usually forgotten when just citing this number.

**Response**: Thanks for this point. We found more related references about the proportion of global GHGs emissions contributed by urban areas. It has been claimed that cities are responsible for as

much as 75% of the GHGs released into the atmosphere (International Energy Agency, 2008; UN-Habitat, 2011), but indeed this proportion of GHGs can be overstated, since it was estimated at about 30~40% when the responsibility for the GHG emissions is allocated based on the production activities, and a larger proportion would be derived when the emissions were assigned to the consumers (Satterthwaite, 2008). This has been noted when citing this number in the introduction (section 1, P2, L11-16).

P2, L6: "few exception such as (Gurney et al, 2012)" -> " few exceptions such as Gurney et al. (2012)". There are actually many other locations where the reference is not properly formatted.

**Response**: The format of references have been carefully checked and revised when needed.

P2, L8: "spatial explicit" -> "spatially explicit"

**Response**: Done.

P2, L15: "Inverse modelling, or top-down approach assimilate" -> "Inverse modelling, often referred to as top-down approach, assimilates"

**Response**: Done.

P2, L19: "by inversion method" -> "by inversion methods"

**Response**: Done.

P2, L26: "are detected" -> "have been detected"

**Response**: Done.

P2, L31: "background area" -> "background areas"

**Response**: Done.

P3, L10: Change to ".. have been identified as major sources of .."

**Response**: Done.

P3, paragraph 2: One more important challenge needs to be added, namely the temporal variation of emissions at diurnal to seasonal time scales, which can not be fully captured by a satellite leading to potential biases in the emission estimates (both to diurnal and seasonal sampling biases).

**Response**: Thanks for this comment. Actually the sampling bias had been mentioned in the discussion. In the revision we have added this point in the introduction as one of the challenges of

detecting urban fossil fuel emissions from space. The corresponding discussion in section 4.3 has also been revised.

P3, L33: "is referred to" -> "was referred to"

**Response**: Done.

Section 2.2.2: It needs to be mentioned here that no temporal variability of emissions was considered. Furthermore, it should also be mentioned that all CO2 was released at the surface or, if not, how emissions were distributed vertically.

**Response**: Thanks. These points have been added in the section 2.4.1 for emission data information. The temporal variability and sampling bias of the satellite data have been mentioned in section 4.3 as a limitation of the OSSEs results.

P6, L16: "using ensemble" -> "using an ensemble

**Response**: Done.

P6, line 29: "an d o" -> "and o"

**Response**: Done.

P6, line 32: What do you mean by "artificially"? This doesn't seem to be the right word here.

**Response**: We wanted to note that the emission optimized by using the integrated ffX$_{CO2}$ enhancements along a latitude range is not so sensitive to transport error in wind direction, which often lead to mismatch of the observed and simulated peaks. We have described this point directly in section 2.5.2 (P11, L4-7).

P7, L19: "the diffusions of fossil-fuel CO2 are" ->" the diffusion of fossil fuel CO2 is"

**Response**: Done.

P7, L26: "in urban canopy" -> "in the urban canopy"

**Response**: Done.

P7, L33: "Los Angeles are used" -> "Los Angeles were used"

**Response**: Done.

P7, L34: "observations are derived" -> "observations were derived"

**Response**: Done.

P8, L22: "is the approximately overpassing time" -> "is the approximate overpassing time"

**Response**: Done.

P9, L33: "with a stronger northern" -> "with a too strong northern" (sounds better to me)

**Response**: Done.

P10, L3: references appear twice

**Response**: Done.

P10, L12: "characterized with" -> "characterized by"

**Response**: Done.

P11, L6: "and Cairo linear" -> "and Cairo a linear"

**Response**: Done.

P12, L13: "this suggest" -> "this suggests"

**Response**: Done.

P12, L19: Please add CarbonSat (Buchwitz, M., M. Reuter, H. Bovensmann, D. Pillai, J. Heymann, O. Schneising, V. Rozanov, T. Krings, J. P. Burrows, H. Boesch, C. Gerbig, Y. Meijer, and A. Loescher, Carbon Monitoring Satellite (CarbonSat): assessment of atmospheric CO2 and CH4 retrieval errors by error parameterization, Atmos. Meas. Tech., 6, 3477-3500, 2013).

**Response**: The reference has been added. Please see P3, L36 and P20, L11.

P13, L18: "across in urban" -> "across the urban"

**Response**: Done.

P14, L13: "using Monte Carlo" ->"using a Monte Carlo"

**Response**: Done.

P14, L25: "Similar magnitude" -> "A similar magnitude"

**Response**: Done.